# NIPMAP: niche-phenotype mapping of multiplex histology data by community ecology

Anissa El Marrahi [1,2], Fabio Lipreri [1,2], Ziqi Kang [1,2], Louise Gsell [1,2], Alper Eroglu [1,2], David Alber [1,2] & Jean Hausser [1,2] ✉

Advances in multiplex histology allow surveying millions of cells, dozens of cell types, and up to thousands of phenotypes within the spatial context of tissue sections. This leads to a combinatorial challenge in (a) summarizing the cellular and phenotypic architecture of tissues and (b) identifying phenotypes with interesting spatial architecture. To address this, we combine ideas from community ecology and machine learning into niche-phenotype mapping (NIPMAP). NIPMAP takes advantage of geometric constraints on local cellular composition imposed by the niche structure of tissues in order to automatically segment tissue sections into niches and their interfaces. Projecting phenotypes on niches and their interfaces identifies previously-reported and previously-unreported spatially-driven phenotypes, concisely summarizes the phenotypic architecture of tissues, and reveals fundamental properties of tissue architecture. NIPMAP is applicable to both protein and RNA multiplex histology of healthy and diseased tissue. An open-source R/Python package implements NIPMAP.

The function of healthy tissues and their disruption in disease depends on the cooperation between cells of different types: hepatocytes in the liver, neurons in the nervous system, immune cells, endothelial cells, fibroblasts, and more[1].

To carry out their functions, cells adopt different phenotypes such as activated or quiescent, adhesive or motile, proliferative or senescent[2]. According to the histological principle of functional zonation, cell types, and their phenotypes organize spatially to facilitate tissue function[3]. For example, in the liver, hepatocytes perform different functions depending on their position along an artery-vein axis[3]. In the lymph node, B cells need to relocalize from the B-cell zone to T-cell zone to potentiate antibody-mediated immunity[4].

The disruption of this organization can directly contribute to disease progression and guide clinical decisions. For example, disruption of pancreatic islet architecture through influx of T cells correlates with the onset of type 1 diabetes[5]. In cancer, tumors can be stratified by the density and distribution of cytotoxic T cells[6]: tumors with dense and uniform T cell infiltration respond best to immune checkpoint inhibitors whereas tumors in which T cells are segregated away from cancer cells show poorer response[6]. Thus, tissue biology and medicine can benefit from characterizing the spatial organization of cell types and their phenotypes in tissues.

In revealing the spatial organization of cells and their phenotypes in tissues, classical techniques such as histology and immunofluorescence imaging are limited to a handful of molecular markers. But in recent years, advances in mass spectrometry such as Multiplexed Ion Beam Imaging (MIBI)[7] and Imaging CyTOF[8,9] have allowed the quantification of dozens of protein and non-protein markers with single-cell resolution. Doing so has also become feasible by multiplex immunofluorescence microscopy thanks to protein-based methods such as t-CycIF[10], 4i[11], Codex[12] as well as RNA-based methods such as MERFISH[13] and in situ sequencing[14,15]. After image processing and cell segmentation, multiplex histology produces rich data in the form of a cell-by-feature table, where features represent the cells' 2D position in the sample, their types, and quantification of dozens to thousands of molecular markers (Supplementary Fig. 1). Markers are typically chosen to identify the type of cells—hepatocytes, neurons, immune, endothelial cells,—and their phenotype.

[1]Department of Cell and Molecular Biology, Karolinska Institutet, Stockholm 171 77, Sweden. [2]SciLifeLab; Solna, Stockholm 171 65, Sweden. ✉e-mail: jean.hausser@scilifelab.se

The recent increase in the number of cell types and phenotypes that can be surveyed in tissue sections leads to a combinatorial challenge in interpreting the data. For example, visualizing the spatial architecture of cellular phenotypes in a multiplex histology dataset of 15 cell types with 15 phenotypic markers in 40 samples requires surveying 9000 images (15 cell types × 15 markers × 40 samples), each with 10,000 to 1,000,000 cells depending on the imaging technology.

Identifying spatial phenotypic interactions—for example, what cancer phenotypes associate with local suppression of anti-cancer immune activity—is even more daunting: co-visualizing all possible pairs of 15 phenotypes for 15 cell types produces 50,000 images (15 cell types × 15 markers to the square) from a single tissue section. An additional layer of complexity is that phenotypes may only interact in specific tissue regions, or at the interface between specific histological niches. To address these combinatorial challenges, a systematic approach is needed to summarize the cellular and phenotypic architecture of tissues and identify its most salient features.

Tissues are structured into histological niches[16]. Within each niche, each cell type has a specific density, defined as the abundance of cells of that type per surface area of the niche. The niche recurs over the tissue section so that a limited number of niches is sufficient to capture the tissue's cellular architecture, by piecing niches together. In this view, interpreting the tissue architecture from multiplex histology data consists in (a) identifying histological niches and (b) segmenting the image into these niches (Fig. 1a).

To automatically identify histological niches, one can determine the local cellular composition at numerous sampling sites—defined as cells found within tissue areas of a given size—across a tissue section. Alternatively, sampling sites can be groups of contiguous cells, identified by graph-based community methods[17]. The histological niches of the tissue are then revealed as clusters of sites with similar cellular composition[17,18].

While the clustering approach has found numerous successful applications in interpreting multiplex histology data so far[17,18],

interpreting the data can benefit from ideas from the field of community ecology, a field with a long history of uncovering spatial patterns[19,20].

Community ecology studies how different species—the ecological analog to our cell types—co-habit in different spatial niches—our histological niches. Within each niche, sites are selected and fieldwork is performed to quantify the species composition at these sites, similar to how multiplex histology surveys cellular composition across a tissue section.

Up until the 1950s, sites and species were then clustered to reveal the organization of species in the different niches (clusters of sites on Fig. 1b). But since the work of Goodall[19], sites are scattered on axes that represent cellular composition using mathematical procedures such as principal components analysis (PCA), so that the proximity of sites reflects the similarity of their cellular composition. Positioning sites on axes of cellular composition allows examining if clustering sites is justified. In the eventuality that sites do not form clusters, the revealed structure of sites can suggest interpretations that better suit the data than clusters (Fig. 1c).

Applying the community ecology approach to multiplex histology data reveals a caveat of clustering sites to identify niches. Sites will form clusters of cellular composition if histological niches occupy distinct areas of the tissue with few interfaces, so that sites belong to only one niche (Fig. 1b). But, when niches colocalize and form larger interfaces, many sites lie at the interface of niches. Because the cellular composition of these sites is a mix (a weighted average) of the corresponding niches, no clear clusters can be distinguished by scattering the cellular composition of sites (Fig. 1c).

Instead, in the case of a two-niche tissue, sites describe a segment in cellular composition space. At each extremity of the segment, we find sites located in the core of the corresponding niche (Fig. 1c). Sites located at the interface between two niches fall in the middle of the segment.

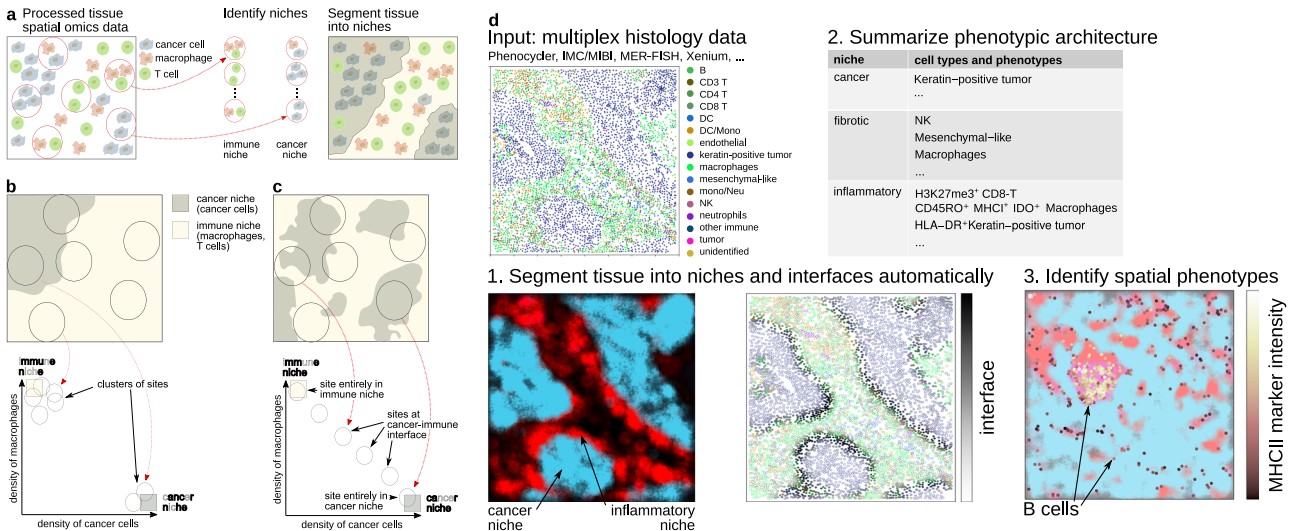

**Fig. 1 | Ideas from community ecology can complement clustering-based approaches in interpreting multiplex histology data. a–c** Clustering can identify the cellular and phenotypic architecture of tissues from multiplex histology data but can misinterpret the niche-interface structure of tissues. **a** To segment histological sections into niches, local cellular composition is surveyed at multiple sampling sites. Sites with similar cellular composition are clustered, and each cluster is interpreted as a histological niche. **b–c** Local cellular composition does not necessarily form clusters. **b** In tissues with little interface, sites mainly fall within a certain niche, and their cellular composition thus clusters by niche. **c** When niches feature large interface regions, sites often cover more than a single niche so that site cellular composition is a mix of the two niches. Consequently, in the case of two

niches, the cellular composition of sites describes a linear segment. Sites from the core region of a given niche are found at the extremities of the segment while sites in the middle of the segment represent interface regions. **d** Niche-phenotype mapping uses ideas from community ecology to automatically segment tissues into niches and their interfaces. For instance, the breast cancer tissue section illustrated here segments into a cancer niche and an inflammatory niche, separated by a cancer-inflammatory interface (1). Based on this segmentation, the phenotypic architecture of the tissue is summarized in terms of the cell types—tumor, macrophage, ...—and phenotypes—CD45RO+, IDO+, ...—associated with different niches and their interfaces (2). Finally, the strongest niche-phenotype associations identified in this way are highlighted to support formulating novel hypotheses (3).

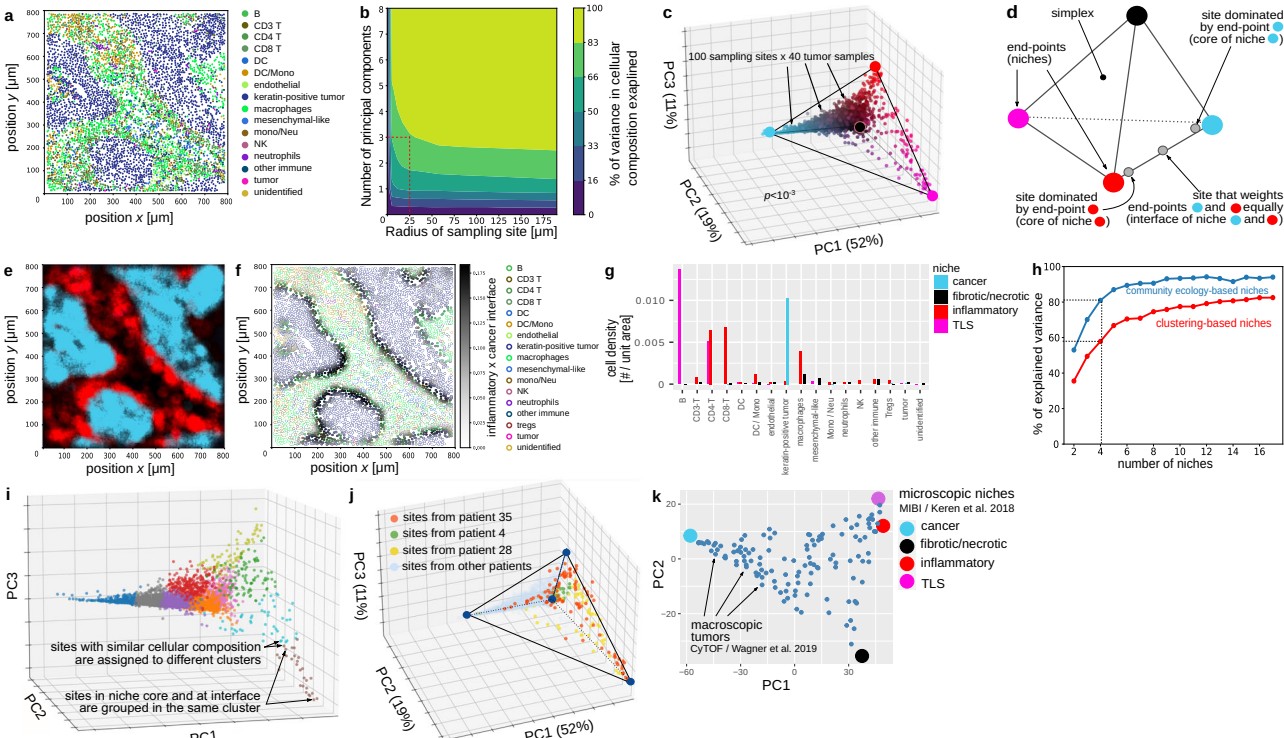

**Fig. 2 | Community ecology-inspired niches offer a quantitative framework to interpret cellular tissue architecture from multiplex histology data. a** MIBI identifies the position and the type of cells in tumor sections. Shown is sample 5 from Keren et al. **b** Co-variance structure emerges when considering the cellular composition of sampling sites 25 μm in size or larger. **c** Representing the cellular composition of sites on three axes (PCs) reveals that site cellular composition does not form clusters but instead describes a continuum constrained by a 3D simplex. P-value: t-ratio test, $n = 4000$ sampling sites, one-sided. **d** The finding that site cellular composition is constrained by a 3D simplex suggests a view of tumor architecture in which local cellular composition is a weighted average of four histological niches, the endpoints of the simplex. Sites located close to an endpoint localize at the core of the niche while sites at the interface between niches fall on the edges of the simplex. **e** Coloring tissue sections according to the local weight of the four niches segments tissue sections into niches. Spatial variations in local cellular composition can be interpreted as space-dependent changes in the weight of the four niches. Colors: local niche weight, as in **c**–**d**. **f** Cells located at interface regions—here the inflammation × cancer interface—are automatically identified as cells where the product of the weights of the inflammation and cancer niches is high. **g** The cellular composition of each niche has a histo-pathological interpretation: cancer, fibrotic/necrotic, inflammatory, and tertiary lymphoid structure (TLS). **h** Community ecology-inspired niches capture tissue cellular architecture more concisely than clustering-based niches. **i.** Community ecology-inspired niches can address artifacts of clustering-based niches. **j.** The same four niches explain both intra-tumor- and inter-patient-variation in the cellular composition of tumors. **k** Niches generalize across breast cancer types and explain inter-patient variation in the macroscopic cellular composition of tumors. DC dendritic cell, Neu neutrophil, Mono monocyte, NK natural killer. Source data are provided as a Source Data file.

The notion that the local cellular composition of tissues does not necessarily form clusters of cellular composition can help interpret multiplex histology data in two ways. First, when sites do not form clusters, many clusters can be needed to describe tissue architecture, potentially leading to an inflation of clusters of unclear histological significance that over-complicate our view of tissue architecture with little scientific benefit. Interpreting local cellular composition as a continuum defined by a parsimonious number of niches can help address this.

Second, interfaces between niches are of interest to interpret tissue dynamics, for example, how tumor progression associates with the biology of the cancer immune interface[6]. Yet, existing approaches to finding interface regions require parameter tuning to specify their cellular composition or the local image properties of interfaces, a time-consuming and potentially subjective process[21–23]. Recognizing that sites that fall in the middle of the segment represent interface regions can identify interfaces automatically.

Here, we implement the community ecology approach for niche-phenotype mapping (NIPMAP, https://github.com/jhausserlab/NIPMAP) of multiplex histology data (Fig. 1d). Applying NIPMAP to protein and RNA multiplex histology of healthy and pathological tissues reveals unexpected geometry in the cellular composition of sites: sites do not form clusters of cellular composition but instead

fall on simplexes, the geometric generalization of triangles or triangular pyramids to arbitrary dimensions. These simplexes are automatically identified using algorithms from satellite image analysis to explain spatial variation in the cellular composition of tissues in terms of histological niches and their interfaces. Projecting cellular phenotypes onto niches and their interfaces reveals known and novel spatial phenotypes, and concisely summarizes how these phenotypes associate with niches and their interfaces (Fig. 1d). Finally, analyzing the niche-interface architecture of tissues uncovered by NIPMAP reveals that (a) spatial context is a stronger determinant of phenotype than cell-autonomous effects, and (b) both niches and their interfaces structure the cellular and phenotypic architecture of tissues (Fig. 1d).

## Results

### Community ecology niches offer a concise and accurate framework to interpret the cellular architecture of tissues

We illustrate the community ecology approach in a multiplex histology dataset of 17 cell types in 40 triple-negative breast tumor samples (Fig. 2a, Supplementary Fig. 1, Methods) from Keren et al.[21].

We determined the cellular composition—the number of cells of each type per unit area (Methods)—of 4000 sites: 100 sites per tumor sample in each of our 40 tumor sections.

Sites are points positioned on axes that represent cellular composition. Because there are 17 cell types, 17 axes (dimensions) are needed in principle. This creates a representation challenge as human intuition is limited to 3 dimensions. However, two principles decrease the number of axes required to interpret tissue architecture. First, the abundance of certain cell types varies little across sites and thus contributes little to tissue architecture so that the corresponding axes can be neglected. Second, the abundance of cell types can correlate across sites—for example, because the cells cooperate in performing a tissue function—so that these cell types can be grouped into a single axis. The axes that optimally capture site cellular composition can be determined automatically by PCA, following the community ecology approach.

To interpret tissue architecture, it is important to set the radius of sampling sites to an appropriate size. Sites need to be large enough to capture local coordination in cellular composition and small enough to avoid blurring this coordination across different niches. To determine an appropriate radius, local coordination was quantified by the number of axes—principal components (PCs)—needed to capture spatial variation in cellular composition. When sampling sites are too small, they cover only one cell at a time: there is little covariance in the cellular composition of sites, and many axes (PCs) are thus needed to capture the cellular composition of sites. Increasing the radius of sites to include neighboring cells reveals covariance structures so that a smaller number of axes (PCs) is sufficient in capturing site cellular composition.

We found that 8 or more PCs are required to capture site cellular composition when the site radius is smaller than 10 µm (-1 cell, Fig. 2b). When sites have a 25µm radius, three PCs are enough to capture 82% of the variance in site cellular composition (Fig. 2b, c). A site radius of 25µm implies that cellular coordination emerges at a length scale of 2–4 cells. Increasing the site radius beyond 25 µm uncovered little novel covariance. We thus set the sampling site radius to 25 µm.

Scattering sites on three PCs revealed no clear clusters (Fig. 2c). Instead, sites described a continuum with the shape of a 3D simplex: a pyramid with a triangular basis. This observation has significance for interpreting tissue architecture. Any point within a simplex can be described as a weighted average of the endpoints that define the simplex (Fig. 2d). Thus, observing that sites are constrained by the geometry of a 3D simplex implies that local cellular composition of the tissue is a mix (weighted average) of four histological niches, the endpoints of the 3D simplex (Fig. 2d). Sites close to endpoints represent cores within the niches, whereas sites halfway between endpoints localize at the interface between two niches (Fig. 2d). This interpretation generalizes the two-niches-and-interface interpretation of continua of site composition introduced in Fig. 1c to more than two niches.

Coloring sites according to their position in the simplex—the contribution of the four niches to site cellular composition—segments tissue sections into histological niches (Fig. 2e). In this niche view of tissue architecture, spatial variation in cellular composition is explained by a locally varying mix of the four niches (Fig. 2e, Supplementary Fig. 2).

The simplex endpoints—and thus the niches—can be identified automatically using hyperspectral unmixing algorithms from the field of satellite imaging[24] or by archetype analysis from machine learning[25,26]. We used the latter algorithms in the present analysis (Methods). The statistical significance of fitting a simplex to sites was quantified using the t-ratio test[27,28] (here $p < 0.001$, $n = 4000$ sampling sites, one-sided, t-ratio = 1.045 in original data, t-ratio 95% confidence interval in shuffled data [1.21–1.56]).

To automatically delineate interface regions between any pair of niches, we note that sites located at the interface between two niches have high weights for both niches. Thus, multiplying the weights of these niches specifically produces high scores for sites located at the interface between the two niches (Fig. 2f). Sorting samples by increasing the prevalence of tumor-immune interfaces (Methods) recovered the mixed vs compartmentalized sample classification

proposed by Keren et al. as well as previously reported differences in the immuno-signaling environment of mixed vs compartmentalized samples (Supplementary Fig. 3a-b).

Examining the niches' cellular composition allows us to interpret the biology of each niche (Fig. 2g). The light blue niche is characterized by a high density of cancer cells (Supplementary Fig. 4a). The black niche features mostly macrophages and mesenchymal-like cells at low density (Fig. 2g) and could thus represent the fibrotic, necrotic niche. In the red niche, we find a mix of CD68+ and MHCII+ macrophages (Supplementary Fig. 4b), CD8 and CD4-T cells, and natural killer (NK) cells (Fig. 2g). A regulatory T cell phenotype can be excluded for the CD4-T cells in this niche as T regulatory cells were assigned their own cell type-based on co-expression of CD4 and FOXP3[21] (Supplementary Fig. 4c). This combination of cell types suggests a type 1 inflammatory region whose function is to trigger anti-cancer immunity[4]. The pink niche may represent the tertiary lymphoid structure (TLS): we find MHCII+ and CD45RO+ B cells and CD4-T cells (Fig. 2g, Supplementary Fig. 4d, e). MHCII and CD45RO are both expressed by B cells activated by antigen recognition[29,30].

We note that niches were determined by collecting 100 sites per sample, so that the total area covered by sites represents 30% of the image area. Such a sampling intensity is sufficient to accurately identify niches while speeding up computations (Supplementary Fig. 5a, b, Methods).

While the four niches correspond to known histological areas from breast pathology[31], clustering-based niches often find a dozen histological niches[8,9,18]. More clusters potentially allow a more accurate description of tissue architecture, at the cost of increased complexity. To determine if increasing the number of niches improves the accuracy of tissue description, we quantified how accurately different numbers of clustering- and community ecology-based niches captured site cellular composition (Methods). We find that 4 community ecology-based niches capture 82% of the inter-site variance in cellular composition while 4 clustering-based niches capture 58% of the variance (Fig. 2h). To describe tissue architecture as accurately as 4 community ecology-based niches, >15 clustering-based niches are needed (Fig. 2h). Thus, community ecology-inspired histological niches provide an accurate yet concise description of tissue architecture.

The community ecology approach can also address artifacts of clustering. For example, sites with similar cellular composition can be assigned to different clusters (Fig. 2i), and a given cluster can contain sites both in the niche core and at its interface (Fig. 2i).

Decreasing or increasing the number of niches from 4 niches down to 2 niches or up to 7 niches causes niches to merge into more coarse-grained niches or split into increasingly fine-grained sub-niches (Supplementary Fig. 6a, b). While we used four niches here to balance accuracy and conciseness, this balance can be tuned by adjusting the number of the niches up or down to zoom in or out on the complexity of tissue architecture.

The number of identifiable niches depends on their prevalence and the amount of available tissue data. A power analysis based on tissue simulations suggests that a single MIBI image is sufficient to capture a niche that covers 3% of the tissue area or more (Methods, Supplementary Fig. 7a, b). The probability of capturing a rare niche scales as the product of niche prevalence times data size, so that increasing the amount of data allows identifying rarer niches. For example, the tissue area of 6 MIBI images allows the identification of niches that occupy less than 1% of the tissue area (Supplementary Fig. 7b).

## Niches identified by NIPMAP generalize across patients of different breast cancer types and connect the microscopic and macroscopic levels of tumor architecture

The four niches are shared across tissue sections. Certain tissue sections make use of all niches (for example patient 35 in Fig. 2j) while others use only a few niches (patient 4 for example, Fig. 2j).

The observation that tissue sections from different cancer patients are composed of the same niches allows for connecting the microscopic cellular architecture of tumors—revealed by multiplex histology—with their macroscopic cellular architecture—revealed by non-spatial methods such as flow cytometry or single-cell mass cytometry. If tumors from different patients are made of the same niches, inter-patient variation in the macroscopic cellular composition of tumors is expected to fall on a simplex whose endpoints represent the four niches (Methods, Supplementary Note 1).

To test this prediction, we compared the macroscopic cellular composition of 128 breast tumors measured by CyTOF[32] with the microscopic niches found in the multiplex histology data of 40 triple-negative tumors from Keren et al.[21]. The macroscopic data of Wagner et al.[32] represents dissociated samples with a volume of 100 mm³, 7 orders of magnitude larger than the microscopic sampling sites we employed in analyzing the data from Keren et al.[21] ($(25\,\mu m)^3 = 1.6 \times 10^{-5}\,mm^3$). The data from Wagner et al.[32] also originates from different breast cancer types—ER+, PR+, Her2+, and triple-negative: this allows testing if microscopic niches identified in the triple-negative breast tumors of Keren et al.[21] generalize across breast cancer types.

As predicted, inter-patient variation in macroscopic cellular composition falls on a low-dimensional simplex bound by the four microscopic niches (Fig. 2k). Cancer and fibrotic niches occupy their own corner of the simplex, as expected. Unexpectedly, the TLS and inflammatory niches share the same corner despite having different cellular compositions (Fig. 2g). This is because the prevalence of the TLS and inflammatory niches is macroscopically coupled in tumors (Supplementary Note 2). The four niches also explain why certain combinations of cell types are found in tumors and why others can never be observed (Supplementary Note 3).

In summary, community ecology-based histological niches emerge from the local cellular composition of tissues at a length scale of 2–4 cells. Niches can be identified automatically by algorithms from satellite image analysis and machine learning. They have a clear histopathological interpretation and provide a concise yet accurate description of tumor architecture that generalizes across patients, tumor types, and the microscopic-macroscopic levels of tumor architecture. This suggests that community ecology-based niches can provide an objective foundation to interpret tissue architecture.

## Niche-phenotype mapping identifies spatial phenotypes and summarizes the phenotypic architecture of tissues

Having identified histological niches and segmented tissue sections accordingly, we determined how niches and their interfaces are associated with cellular phenotypes.

To do so, we took advantage of single-cell, spatial measurements of 18 phenotypic markers also profiled by Keren et al.[21] alongside the 17 lineage markers used to determine cell types (Supplementary Fig. 8a). We looked for phenotypic markers whose intensity associates with the position of cells in a given niche or at a given interface. The position of cells in a niche was quantified as the weight of that niche. Similarly, the position of cells at an interface between two niches was quantified as the product of the weights of these two niches. We then correlated the niche/interface weight with the intensity of phenotypic markers (Spearman's rank order correlation coefficient $\rho$, $p$ values: two-sided $t$ test, false discovery rates from Strimmer[33], Methods).

Statistically significant niche-phenotypes associations (fdr < 1%, $\rho > 0.3$) were ordered by cell types and visualized as a heatmap (Fig. 3a). To explore these associations in a phenotype-centered rather than in a cell type-centered manner, niche-phenotypes associations can also be sorted by phenotypes (Supplementary Fig. 8b). Out of the 3040 possible niche-phenotypes associations in this dataset (16 cell types × 19 phenotypic markers × 10 niches and interfaces), significant associations were reported as a table which concisely summarizes the phenotypic architecture of the tissue (Table 1).

Niche-phenotype mapping recovered expected spatial phenotypes. For example, among B cells, the HLA-DR (MHCII) phenotype is associated with the TLS niche while HLA-DR negative cells—presumably plasma cells—localize in other niches (Fig. 3b)[4]. Neutrophils and tumor cells with an HLA-DR (MHCII) phenotype localized in the inflammatory region (Fig. 3c, Table 1). While MHCII expression is normally restricted to antigen-presenting cells of the immune system[4]—dendritic cells, macrophages, B cells—MHCII+ neutrophils are emerging as actors in anti-tumor immunity[34]. MHCII expression in tumor cells has also been reported previously and associates with positive prognosis[35].

In keratin-positive tumor cells, the MHCI marker associated with the interface of cancer and inflammation niches (Fig. 3d, Supplementary Fig. 10). This suggests that MHCI expression in tumor cells could determine the position of the cancer-inflammation interface. Alternatively, the proximity of the inflammatory niche could induce MHCI in neighboring cancer cells or secrete MHCI as a soluble form ref. [36] (Supplementary Fig. 10).

Niche-phenotype mapping also highlighted unexpected spatial phenotypes. CD45RO+ macrophages and dendritic cells localized in the inflammatory niche (Fig. 3e, Supplementary Fig. 9). CD45, a commonly used marker of bone marrow-derived immune cells, has several splicing isoforms. The CD45RO isoform is a marker of activated and memory T cells as well as activated B cells with highly mutated B-cell receptors (BCR)[29,30]. In the context of macrophages, CD45RO was previously reported to be the dominant CD45 isoform and to function as a cell adhesion receptor that inhibits pro-inflammatory macrophages[37]. The literature suggests that the CD45RO signal we analyze here is specific to the CD45RO isoform (see Discussion).

Keratin 6, a highly abundant protein that forms intermediate filaments in the cytoskeleton of epithelial cells[38] was found in dendritic cells of the inflammatory niche (Fig. 3f), perhaps because DCs first migrate to the cancer niche where they take up Keratin 6, before re-localizing to the inflammatory niche.

Neutrophils located at the interface of cancer and inflammatory niches are positive for the immunosuppressive markers IDO and PD-L1 (Fig. 3a, g, Table 1). This suggests a potential role of neutrophils in facilitating the immune escape of cancer cells.

## Both niches and their interfaces structure spatial variation in cellular phenotypes

NIPMAP can be used to explore fundamental questions regarding the cellular and phenotypic architecture of tissues.

We illustrate this by examining the origin of phenotypic heterogeneity of the tumor microenvironment. This heterogeneity could originate from the spatial context of cells, with local signaling cues determining cellular phenotypes. Phenotypic heterogeneity could also stem from cell-autonomous phenomena such as catching cells at different points of differentiation trajectories or from stochasticity in adopting different phenotypes. Cell-autonomous and spatial contexts can both contribute to phenotypic heterogeneity[39–42].

Understanding the relative contribution of cell-autonomous phenomena vs spatial context to phenotypic heterogeneity has practical implications for spatial and single-cell omics data analysis. If spatial context drives phenotypes, phenotypes relevant to spatial tissue architecture are expected to emerge in spatially-agnostic analyses such as phenotypic clustering[17]. But spatially-agnostic methods are expected to struggle at identifying spatial phenotypes if cell-autonomous factors dominate phenotypic heterogeneity: in this case, spatial approaches to spatial phenotype identification such as NIPMAP are needed.

We find that niche-phenotype mapping and (spatially-agnostic) phenotypic clusters identify shared as well as different phenotypic markers (Fig. 4a for dendritic cells, other cell types in Supplementary Fig. 8c–e). For example, in DCs, the CD45RO and Keratin 6 markers

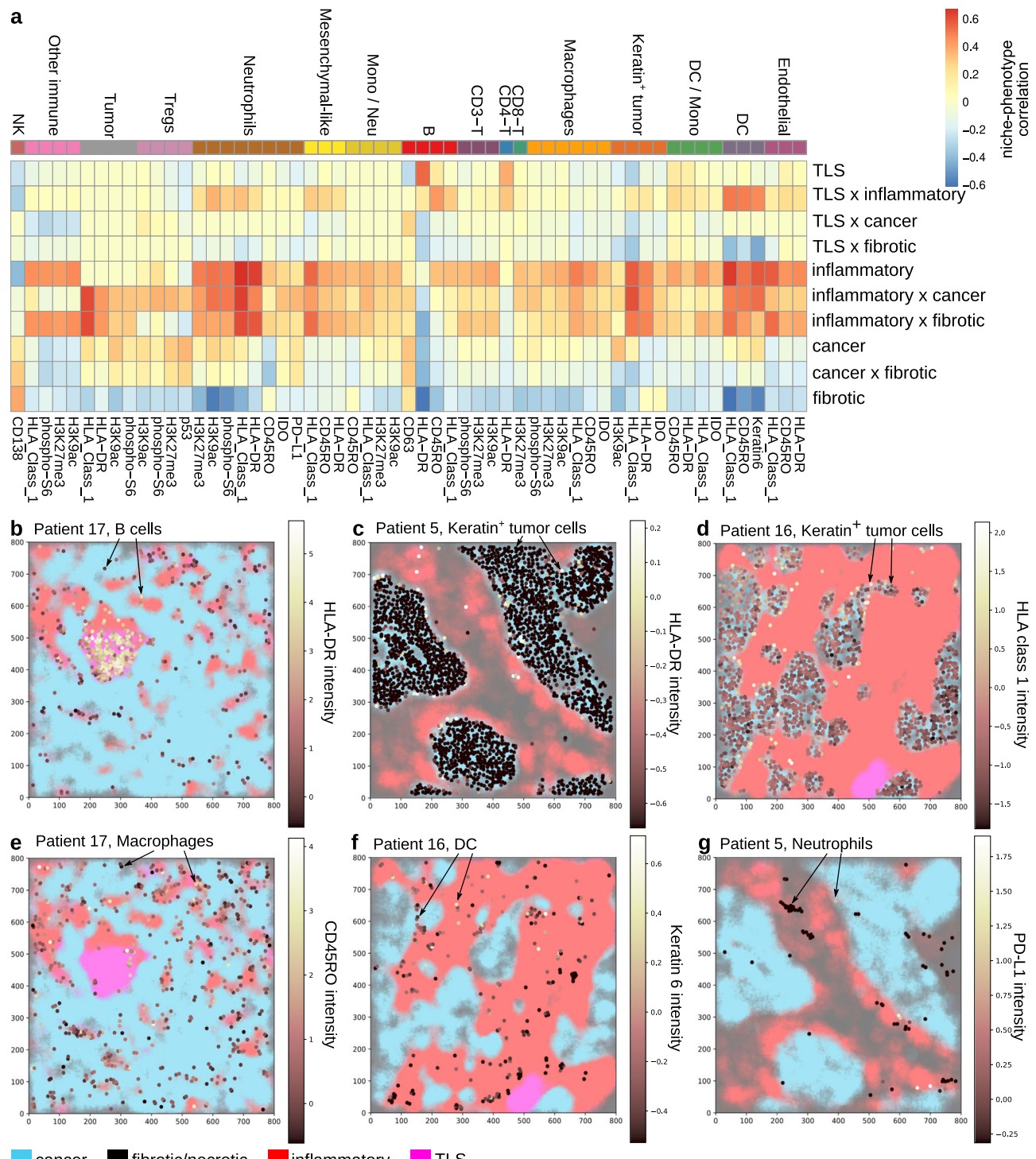

**Fig. 3 | Niche-phenotype mapping identifies spatial phenotypes and summarizes the phenotypic architecture of breast tumors. a** Different cell types have phenotypes (columns) with specific associations to the different histological niches and their interfaces (rows). Shown are all phenotypes with a niche-phenotype correlation of at least 0.3 and a false discovery rate of less than 1%. **b**–**g** Niche-phenotype mapping reveals expected (**b**–**d**) and novel (**e**–**g**) niche-phenotype associations, which are visualized by overlaying phenotypic marker intensity (dot color) and tissue segmentation (background color). Cells of types other than the type indicated in each panel are not shown for clarity. **b** B cells are positive for HLA-DR in the tertiary lymphoid structure (TLS) niche. **c** Keratin-positive tumor cells are positive for HLA-DR/MHC class II in the inflammatory niche. **d** Keratin-positive tumor cells are positive for HLA class I at the interface of the cancer and inflammatory niches. **e** CD45RO intensity in macrophages is associated with the inflammatory niche. **f** Keratin 6 in dendritic cells is associated with the inflammatory niche. **g** PD-L1 intensity in neutrophils associated with the interface of cancer and inflammatory niches. Marker intensity represents $Z$ scored marker abundance, as quantified in the original study[21]. Source data are provided as a Source Data file.

associate with the inflammatory niche and define phenotypic clusters 4 6 and 1. Yet, niche-phenotype mapping identifies an additional marker, HLA class 1, not highlighted by phenotypic clustering. This suggests that clustering can fail to highlight spatial markers, perhaps because phenotypic clusters are not just driven by space but also by cell-autonomous effects independent of spatial context. In support of this hypothesis, phenotypic clustering highlights CD138 and HLA-DR, both of which are poorly associated with space in DCs (Fig. 4a).

This raises the question of the relative contribution of cell-autonomous phenomena and spatial context to the phenotypic heterogeneity of the tumor microenvironment.

If cell-autonomous effects dominate phenotypic heterogeneity, phenotypic clusters are expected to show poor association to space: clustering cells by phenotypic markers without regard to spatial context will produce clusters that poorly predict a cell's niche (Fig. 4b) Conversely, phenotypic clusters will predict the niche if spatial context determines phenotypic heterogeneity (Fig. 4c).

To test this, we examined how tightly phenotypic clusters associate with spatial context. As an upper bound for how precisely the marker panel and marker quantification can position cells in space, we use a linear predictor of a cell's niche from phenotypic marker intensities (area under the curve = 0.89 in predicting which DCs located in the inflammatory niche, Fig. 4d).

In dendritic cells, we find that one phenotypic cluster predicts the inflammatory niche as accurately as the linear model (Fig. 4d). Other DC clusters predict the location of DCs in other niches (Supplementary Fig. 8c). These observations generalize to other cell types (Supplementary Fig. 8d, e).

This suggests that the phenotypic heterogeneity of the tumor microenvironment is driven both by the spatial context—niche or interface—in which cells find themselves and by cell-autonomous effects, with spatial context playing a bigger role than cell-autonomous effects.

Another fundamental question of tissue architecture is whether niches contribute more to structuring phenotypic heterogeneity than interfaces.

To find out, we quantified how much a given niche or interface associates with phenotypic heterogeneity by summing the squared correlations of all phenotypes of all cell types for that niche or interface (the rows of the matrix in Fig. 3a). If niches structure phenotypic heterogeneity more than interfaces, the 4 niches are expected to have a larger sum of squared correlations compared to the six interfaces (Fig. 4e). Conversely, if interfaces structure phenotypic heterogeneity more than niches, the interfaces are expected to have a larger sum of squared correlations (Fig. 4e).

We find that both niches and interfaces can have a large sum of squared correlations (Fig. 4e). The inflammatory and fibrotic niches as well as the inflammatory-cancer and inflammatory-fibrotic interfaces contribute most to phenotypic heterogeneity in the context of the present phenotypic marker panel. This suggests that phenotypic heterogeneity is structured by both niches and interfaces. Interfaces thus represent histological areas in which cells adopt specific phenotypes, different from the niches that meet at the interface. For example, phospho-S6+ Tregs and phospho-S6+ macrophages are specific to the cancer-inflammatory interface while CD45RO+ B cells specifically located at the TLS-inflammatory interface.

## NIPMAP identifies the cellular and phenotypic architecture of developing lung profiled by in situ RNA sequencing

So far, we applied NIPMAP to spatial profiling of tumor tissues at the protein level. However healthy tissues and RNA profiling data can also be interpreted with NIPMAP. We illustrate this by applying NIPMAP on single-cell, spatial RNA profiling of healthy embryonic human lungs by In Situ Sequencing (ISS, Fig. 5a)[43,44].

**Table 1 | NIPMAP concisely summarizes the cellular and phenotypic architecture of tissue samples as a table of niches/interfaces and associated cellular phenotypes**

| niche | cell phenotype |
|---|---|
| cancer | H3K9ac+ keratin-positive tumor |
| | CD63+ B |
| fibrotic | CD138+ NK |
| | Mesenchymal-like |
| | Macrophages |
| | CD8-T |
| inflammatory | H3K27me3+ CD8-T |
| | CD45RO+ H3K27me3+ H3K9ac+ HLA class 1+ IDO+ Macrophages |
| | HLA-DR+ keratin-positive tumor |
| | CD45RO+ H3K27me3+ H3K9ac+ HLA class 1+ HLA-DR+ Neutrophils |
| | CD45RO+ HLA class 1+ Keratin 6+ DC |
| | CD45RO+ HLA class 1+ HLA-DR+ Endothelial |
| | CD45RO+ HLA class 1+ HLA-DR+ Mesenchymal-like |
| | HLA class 1+ B |
| | CD4-T |
| TLS | HLA-DR+ B |
| | HLA-DR+ CD4-T |
| | CD8-T |
| inflammatory × cancer | HLA class 1+ keratin-positive tumor |
| | phospho-S6+ Macrophages |
| | IDO+ PD-L1+ phospho-S6+ Neutrophils |
| | H3K27me3+ H3K9ac+ phospho-S6+ Tregs |
| | H3K9ac+ HLA class 1+ phospho-S6+ Tumor |
| inflammatory × fibrotic | IDO+ keratin-positive tumor |
| | HLA-DR+Tumor |
| TLS × inflammatory | CD45RO+ B |

The table reports the niche-phenotype associations with a correlation of at least 0.3 and a false discovery rate of at most 1%, as shown in Fig. 3a.

Similar to tumors, we find that covariance structure in local cellular composition emerges in sites with a 25 μm radius (Supplementary Fig. 11a). Four PCs are sufficient to capture 85% of the spatial variation in the cellular composition of the tissue.

Spatial variation in the cellular composition of the developing lung fits a simplex with five endpoints (Supplementary Fig. 11b), suggesting 5 niches ($p = 0.001$, $n = 20,000$ sites, one-sided t-ratio test, t-ratio statistic = 1.496 in original data, t-ratio 95% confidence interval in shuffled data [1.59 – 2.07]). A 5-end-point simplex is a four-dimensional geometrical object which makes it difficult to visualize. To address this, we projected sites on the faces of the simplex (Methods). Examining the distribution of projected sites on the faces of the simplex, we observed sites close to all 5-endpoints, supporting the existence of all 5 niches (Supplementary Fig. 11b).

Quantifying the cellular composition of each niche suggested (1) epithelial, (2) parenchymal, (3) smooth muscle, and (4) vessel niches, as well as (5) ductal and alveolar (liquid-filled) space (Fig. 5b, c). In well-formed ducts, we observed that the epithelium separates the ductal space from the smooth muscle niche (Fig. 5b, d), as expected. The vessel niche does not associate with the alveolar space nor the epithelial niche (Fig. 5b, Supplementary Fig. 11b), as expected at this stage of development (week 13)[44].

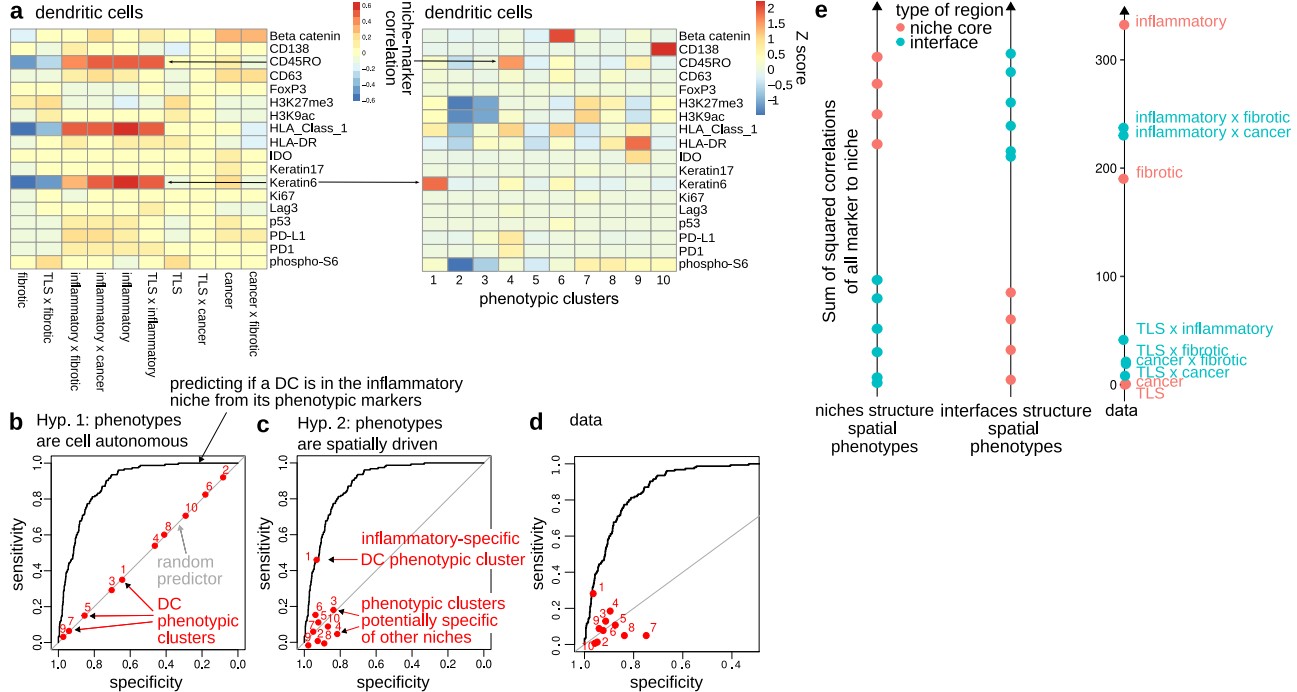

**Fig. 4 | Niche-phenotype mapping reveals two fundamental properties of tissue phenotypic architecture. a**–**d** The spatial context of cells is a stronger determinant of phenotype than cell-autonomous effects. Dendritic cells are illustrated here and other cell types appear in Supplementary Fig. 8d, e. **b** Niche-phenotype mapping (left) and (spatially-agnostic) phenotypic clusters (right) identify common phenotypic markers—CD45RO, Keratin 6—but also divergent markers—HLA class 1, HLA-DR, CD138. **b** If cell-autonomous effects determine phenotypes more than spatial context, spatial context will poorly associate with phenotypes. Thus, phenotypic clusters—which are built independently of spatial context—will predict a cell's niche as poorly as a random predictor. The sensitivity and specificity of phenotypic clusters are computed using cluster membership as a predictor of niche membership. **c** If the spatial niche context of a cell is the main determinant of phenotypes, phenotypes will strongly associate with niches. Thus, the niche of a cell can be predicted from a cell's phenotypic cluster. **d** The data supports the hypothesis that the spatial context of cells is a stronger determinant of phenotype than cell-autonomous effects. **e** Both niches and their interfaces structure the spatial architecture of phenotypes. The contribution of a niche or interface to phenotypic architecture is quantified by summing the squared correlations of all phenotypes with that niche (rows of the matrix shown in Fig. 3a). Squared correlations are expected to be higher for niches and lower for interfaces in a scenario where niches structure phenotypic architecture (left). Conversely, under the hypothesis that phenotypic architecture is structured by interfaces, squared correlations will be higher for interfaces and lower for niches (middle). The data shows that both niches and interfaces have high squared correlations and thus contribute to phenotypic architecture (right). TLS: tertiary lymphoid structure. Source data are provided as a Source Data file.

As in tumors, we find that niches structure phenotypic architecture (Fig. 5e, Supplementary Fig. 11c). For example, localization in the vessel niche of arterial cells and pericytes correlated with expression of *JAG1* (Fig. 5f) whose functions in endothelial development were previously reported[45]. The platelet-derived growth factor A (PDGFA) ligand and receptor respectively are associated with the vascular vs parenchymal context of pericytes (Fig. 5e, Supplementary Fig. 11d, e).

To test the robustness of the identified niches with respect to cell type granularity, we repeated niche identification using the 73 cell types proposed by Sountoulidis et al.[44] instead of the 32 coarser-grained cell types of Fig. 5. Increasing the number of cell types, we find five niches with cellular composition and spatial distribution similar to those found with 32 cell types (Supplementary Fig. 12). This suggests that niches show a degree of robustness to cell type granularity.

Thus NIPMAP generalizes to spatial RNA profiling data and healthy tissues.

## Discussion

Multiplex histology produces rich datasets in the form of the location of 10,000–1,000,000 cells, dozens of cell types, and dozens to thousands of phenotypic markers. 100,000+ images of phenotype interactions can be produced from a single sample, which leads to a combinatorial challenge in visualizing and interpreting the data. To address this, we introduce NIPMAP, adapting methods from community ecology and satellite image analysis to multiplex histology data in order to (a) identify the histological niches underlying spatial tissue

architecture and (b) summarize how histological niches and their interfaces structure cellular phenotypes.

Applying NIPMAP to multiplex histology data from healthy and disease samples reveals that the local cellular composition of tissues has the low-dimensional geometric structure of a simplex. The endpoints and halfway points of the simplex represent histological niches and their interfaces. Niches match known histo-pathological areas and provide a concise yet accurate summary of tissue architecture.

In the context of breast tumors, these niches generalize across patients and tumor types and connect the microscopic and macroscopic levels of cellular architecture. Individual phenotypic markers are mapped on these niches to identify spatial phenotypes and summarize how phenotypes integrate into histological niches and their interfaces. Analyzing how phenotypes associate with niches and their interfaces suggests that spatial context and cell-autonomous effects both determine phenotypes, with the former having a larger influence than the latter. Phenotypic heterogeneity is structured both by niches and their interfaces, with interfaces being home to specific cellular phenotypes.

Errors in cellular segmentation and lateral signal spill-over can lead to mis-assigning cell types and phenotypes, potentially leading to false positives or false negatives during niche-phenotype mapping[46] (see the p53 marker in Supplementary Fig. 10 for example). Even in perfectly segmented tissues, marker signal can be mis-attributed: a marker can associate with cells of a given type within a given niche not because cells of that type express the marker but instead because the

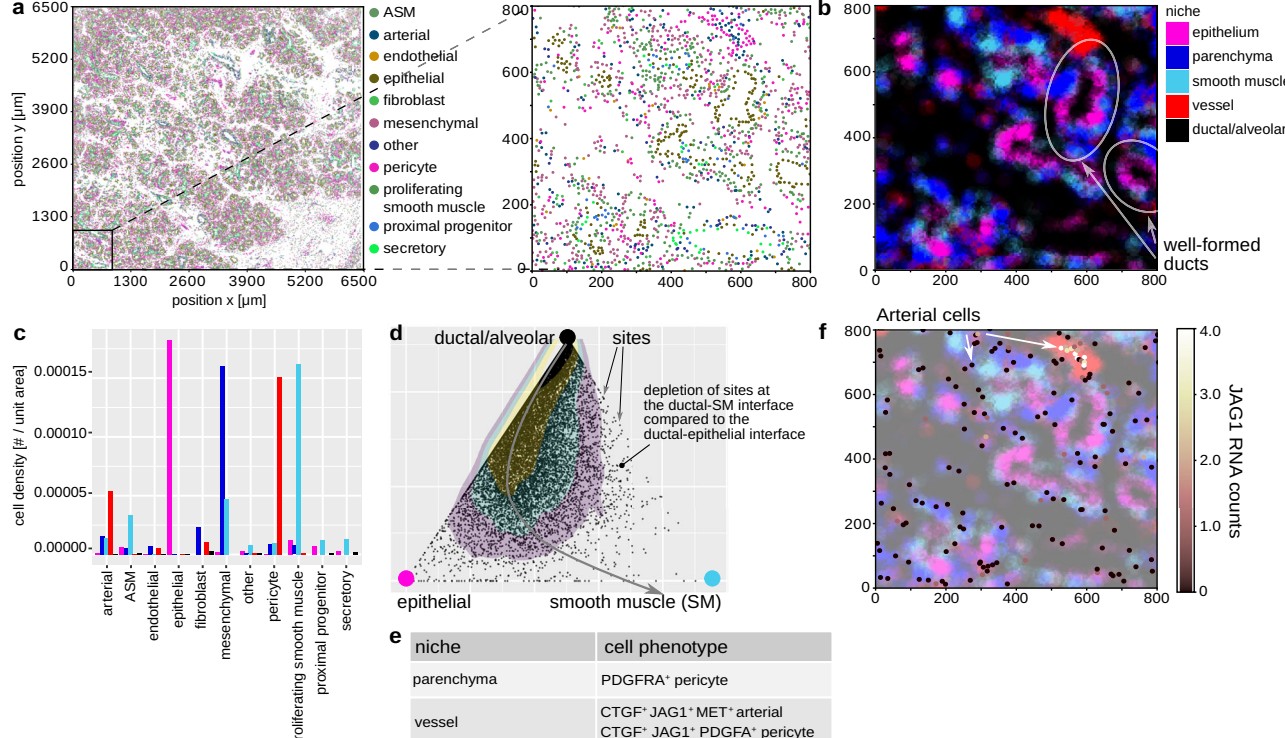

**Fig. 5 | NIPMAP generalizes to RNA-based spatial profiling of healthy tissue.**
**a** The cellular and phenotypic architecture of the human developing lung was
characterized by in situ RNA sequencing. Data: Sountoulidis et al.[44]. **b** The devel-
oping lung can be segmented into 5 histological niches. **c** Each niche has a specific
cellular composition suggesting the histological basis of that niche. **d** Niche-
phenotype mapping recovers known spatial associations between niches, such as
the epithelial niche separating the smooth muscle niche from ductal/alveolar
space. Dots: sites projected on the face of the 5 niches simplex defined by the
ductal/alveolar, epithelial and smooth muscle niches. Colored areas: contours of
dots density. The sequential organization of the ductal, epithelial, and smooth
muscle niches is reflected by a depletion of sites at the ductal-smooth muscle
interface relative to the ductal-epithelial interface. Thus, from the ductal/alveolar
space, we first encounter the epithelial niche. Beyond the epithelial niche, the
smooth muscle niche increases in weight. **e** Projecting phenotypes onto niches
identifies cell types and phenotypes that associate with specific niches. **f** Arterial
cells express *JAG1* when located in the arterial niche but not when located in other
niches. Dots: arterial cells. Color bar: *JAG1* RNA count per cell. Background color:
niche segmentation. ASM airway smooth muscle. Source data are provided as a
Source Data file.

marker is systematically present in that niche, perhaps as a soluble
form or as a constituent of the extra-cellular matrix (see the CD138
marker in Supplementary Fig. 9 for example). NIPMAP is designed as
the final layer of the multiplex histology data processing stack. It does
not attempt to correct cellular segmentation, cell type assignment, or
signal attribution errors: these issues need to be addressed in the
corresponding layers. These issues are recognized in the multiplex
histology field and ongoing methodological research is seeking to
address them[46–50]. The statistical methodology employed by NIPMAP
provides a degree of robustness to segmentation, cell type and signal
attribution errors because niche-phenotype associations are only
captured if they occur systematically across cells. Despite that, and
until a definitive methodological solution to cellular segmentation, cell
type assignment or signal attribution errors is established, spatial
phenotypes highlighted by NIPMAP need to be confirmed by over-
laying cellular segmentation with the spatial signal distribution of the
corresponding phenotypic marker (Supplementary Figs. 9, 10).

NIPMAP identifies novel spatial phenotypes to the best of our
knowledge, such as an association between CD45RO and the locali-
zation of macrophages in the inflammatory niche of triple-negative
breast tumors. CD45 is a hematopoietic marker that can be expressed
as different isoforms CD45RA, CD45RB, and CD45RC depending on
alternative splicing of its three exons A, B, and C. The CD45RO isoform
lacks the A, B, and C exons. The different isoforms are associated with
specific phenotypes, at least in the context of lymphocytes cells[29,30].
Thus, quantifying the CD45 isoforms specifically is a prerequisite to

interpret the observation that CD45RO+ macrophages localize in the
inflammatory niche of triple-negative tumors. The existing literature
suggests that the CD45RO antibody (UCHL1 clone, Biolegend) used in
the MIBI study we re-analyzed here[21] is specific to the CD45RO isoform
of CD45. The UCHL1 clone has been used to specifically quantify
CD45RO since the 1980s[51–54]. An early flow cytometry study in T cells
found that the abundance of CD45RA and CD45RO abundance—as
quantified by UCHL1—correlate negatively, consistent with the speci-
ficity of UCHL1 to the CD45RO isoform[52]. Profiling CD45RO by means
of the UCHL1 clone alongside CD45RA (by means of the 2H4, F8-11-13,
or HI100 clones) is routinely used to discriminate between memory
and naive T cells[52,54]. In addition, the risk of non-specific quantification
of CD45 isoforms in macrophages is mitigated by western blot, flow
cytometry and full-length RNA-seq observations that bone marrow-
derived macrophages from the spleen, white adipose tissue, liver, and
the peritoneal cavity lack the CD45RA, RB and RC isoforms and thus
specifically express the CD45RO isoform[37]. These observations suggest
that macrophages specifically upregulate the CD45RO isoform in the
inflammatory niche of triple-negative breast tumors.

When performing single-cell analyses, a decision needs to be
made regarding the granularity of cell types. For example, T cells could
be lumped together with other immune cells or be assigned a more
granular type such as Th17 CD4+ T cell. Alternatively, cell typing could
have intermediate granularity - such as CD4-T cells—and com-
plemented by phenotypes—such as Th17. This raises the question of
how cell type granularity impacts niche identification. In general, one

expects optimal niche identification when analyzing cell types at a range of intermediate granularities and poor niche identification when the cell types are insufficiently granular or too granular. This is because insufficient granularity can prevent observing the cell types that characterize a given niche: for example, a vascular niche characterized by pericytes and endothelial cells cannot be identified if cells are coarsely grouped into epithelial vs non-epithelial. Conversely, too granular cell types–for example if each cell has its own type - prevent identifying recurrent patterns in the local cell type composition of tissues to reveal its niches. In the context of the ISS data of Sountoulidis et al.[44], our findings suggest that the niches identified are robust to the exact number of cell types used.

To interpret spatial phenotypes, markers need to be separated into (1) markers of cell type and (2) markers of phenotypes, for exclusive use during niche identification and niche-phenotype mapping respectively. This is because using a given marker both for cell typing and phenotyping would necessarily identify the phenotype in the niches where the cell type is present, thus biasing niche-phenotype mapping. Both datasets used here comply with this separation. The MIBI data from Keren et al. used 17 lineage markers to define cell types and another 18 functional markers to identify phenotypes[21]. The ISS data of Sountoulidis et al.[44] used 72 lineage markers to identify cell types and profiled a distinct set of 75 genes from the WNT, SHH, NOTCH, and RTK pathways[44] which we used here in niche-phenotyping mapping.

NIPMAP can complement existing methods to analyze multiplex histology data. For example, cellular spatial enrichment analysis aims at identifying spatial interactions by looking for pairs of cell types that colocalize more often than expected by random chance, defined by shuffling cell positions within a given niche[21]. By providing an approach to identify niches automatically, NIPMAP could facilitate cellular spatial enrichment analysis. Other spatial analyses such as proximity analysis and nearest neighbor analysis could benefit from NIPMAP's niche identification in the same way.

To cluster-based methods aimed at identifying discrete cellular structures in multiplex histology data such as community detection based on spatial cellular networks and cellular neighborhood analysis[17,18], NIPMAP adds two principles. The first is that the number and identity of histological niches can be determined by exploiting the simplex geometry of local cellular composition. The second is that the simplex geometry provides a criterion to distinguish niches from interfaces and automatically identify interfaces without parameter tuning. Identifying interface regions is of high interest to tissue biology. In cancer, for example, understanding why immune cells from the inflammatory niche fail to penetrate the cancer niche can suggest therapeutic interventions to remove blocks to anti-tumor immunity[6].

Here we illustrated NIPMAP on spatial data from MIBI[7], a protein-based approach, and ISS[43], an RNA-based method. Beyond these two technologies, NIPMAP is designed to be applicable to other spatial methods with single-cell resolution such as Imaging CyTOF, Codex, CyclF, 4i, MERFISH, and more[10–13].

Similar to how community ecology defines ecological niches based on local species covariance, local covariance between cell types is exploited by NIPMAP to identify histological niches. This approach requires assigning a type to each cell based on marker intensities and prior knowledge, with two potential downsides. First, niches identifiable in this way could be limited by previously known cell types. Second, assigning types to individual cells from marker intensities is time-consuming and not guaranteed to be error-free due to segmentation errors, signal misattribution, non-specific antibody binding, auto-fluorescence, molecular exchanges between cells, and more. To address this, it would be desirable for niche identification to be based not on the types of cells but instead on marker intensities of local tissue regions prior to segmentation into cells. Preliminary exploration of this question in the context of the MIBI data of Keren et al.[21] suggests

that niches can potentially be identified without assigning predefined types to cells, resulting in similar niches as cell type-based niche identification (Supplementary Note 4). Future research can further develop this methodology.

In the future, the cellular and phenotypic architecture identified by NIPMAP could support efforts aimed at understanding how healthy tissues maintain function despite the need for constant cellular turnover[55] and interpreting the spatial dynamics of niches during histological processes such as development, tissue repair or disease progression[44,56,57].

## Methods

### Ethics statement
MIBI data of TNBC of Keren et al.[21]: all the MIBI samples came from archival tissue blocks housed in the Stanford Pathology tissue bank that were sourced from primary surgical resections. Since no material was acquired prospectively for the study, acquiring MIBI data on these samples was not deemed human subjects research, and requirements for an ethical permit were waived by the institutional review board.

Fetal lung ISS data of Sountoulidis et al.[44]: ethical permit was obtained by the authors of the initial study from the Swedish National Board of Health and Welfare. The analysis was approved by the Swedish Ethical Review Authority (2018/769-31). The clinical staff of the initial study acquired informed written consent from the donor.

CyTOF data of Wagner et al.[32]: tissue were collected after obtaining written informed consent from patients at the University Hospital Basel (Switzerland), the University Hospital Zurich (Switzerland), and in collaboration with the Patient's Tumor Bank of Hope (PATH, Germany) at the breast cancer centers at St. Johannes Hospital Dortmund and Institute of Pathology at Josefshaus (Germany) and the University Hospital Giessen and Marburg, Marburg site (Germany). The collection was approved by the Ethics Committee Northwest/Central Switzerland (#2016-00067), the Ethics Committee Zurich (#2016-00215), and the Faculty of Medicine Ethics Committee at Friedrich-Wilhelms-University Bonn (#255/06).

We did not carry out sex/gender analysis for two reasons. First, the experimental unit of our study is cells and groups of cells, not human individuals. Second, the aim of our study introduce a computational method that detects spatial patterns of cellular organization in tissue. Such patterns of cellular organization are a hallmark of multi-cellular organisms, including humans, across biological (sex, age) and social groups (genders, ethnicity). Examining if and how specific details in these spatial patterns vary across biological and social groups is beyond the scope of the present study.

### NIPMAP methodology overview
Sites are sampled from the tissue and their composition in terms of (predetermined) cell types is estimated with a Gaussian kernel. The main co-variance axes of cellular composition are identified by PCs analysis. Archetype analysis is used to fit a simplex to site cellular composition and thereby identify histological niches. From these niches, the original tissue is spatially segmented into niches and interface regions. Niches and interfaces are associated with phenotypic markers by correlation analysis to (1) summarize tissue phenotypic architecture and (2) identify salient spatial phenotypes.

### Processing the MIBI data from Keren et al
From the website of the Angelo lab, we obtained processed MIBI data for 36 protein markers from 41 TNBC patient samples: intensity values, segmented images, and patient data. The 41 samples represented patients aged 26-91 (mean 54.2 years). While gender information was not included, samples are expected to be female because 99% of breast tumor patients are female.

The segmented data (cellData.csv) contained $(x, y)$ coordinates of each cell and its type (out of 17 cell types) as determined by the authors

of the study. Following the authors of Keren et al.[21], patient 30 was excluded from the analysis.

## Quantifying cell type density in sites with a Gaussian kernel

Each tissue slide is a 2-dimensional space with cells of a determined cell type as points of coordinates $(x, y)$. We positioned 100 sampling sites randomly on each slide, by drawing the centers from a uniform distribution. In contrast to common practice, sites are not positioned on cells but uniformly across the tissue section. Doing so has the advantage that sites are representative of tissue architecture and unbiased by spatial variation in cellular density.

We generated 4000 sites, 100 sites for each of the 40 slides.

To quantify the abundance of cells of different types at each site, rather than counting cells with a circle of radius $r$, we used a Gaussian kernel density estimation to decrease counting noise: if a cell is slightly outside the circle of radius $r$, it is not counting with the first strategy. Counts are thus sensitive to slight changes in the position of the center of the circle. Gaussian kernel density estimation addresses this by weighting cell counts by their distance to the center in a smooth fashion. The weight $g$ of a cell of position $\boldsymbol{x}$ (with $\boldsymbol{x}$ a vector) to the site of center $\boldsymbol{s_k}$ is defined as

$$g(\boldsymbol{x}, \boldsymbol{s_k}) = \frac{1}{2\pi r} e^{-\frac{1}{2}\left(\frac{\|\boldsymbol{s_k} - \boldsymbol{x}\|}{r}\right)^2} \qquad (1)$$

We summed up the density values for each cell type.

We performed PCA, centered and unscaled using the ade4 package of the data analysis software R[20].

In positioning sites, we excluded areas of the slides located within distance $r$ from the image edge, in order to decrease edge effects.

We explored a broad range of width $r$ values to examine the robustness of tumor architecture (Fig. 2b). We found that $r = 25\ \mu m$ is the minimal radius allowing to capture cellular architecture (Fig. 2b). This suggests that tumor micro-architecture emerges on a scale of 2-4 cells.

## Archetype analysis

Archetype analysis[25] aims to fit a $d$-dimensional simplex as tightly as possible to $n$ data points $\boldsymbol{x}$. The simplex has $p$ endpoints $\boldsymbol{b_k} \in \mathbb{R}^d, k = 1, \ldots, p$ which represent the endpoints, also known as archetypes[26]. By definition, each point $\boldsymbol{x}$ within the simplex can be written as a weighted average of the endpoints

$$\boldsymbol{x} = \sum_{k=1}^{p} \alpha_k \boldsymbol{b_k} \qquad (2)$$

with the weights $\alpha$ constrained by $0 \le \alpha_k \le 1$ and $\sum_{k=1}^{p} \alpha_k = 1$. We used the tumor samples projected onto the 3 first PCs as input for archetype analysis. We used the Archetypal Analysis python package[58], with the parameters: n_archetypes = 4, tolerance = 0.001, max_iter = 200, random_state = 0, C = 0.0001, initialize = 'random', redundancy_try = 30. The output of this algorithm contains a dataset of $\alpha_k$ weights for each tumor sample and the coordinates of the endpoints $\boldsymbol{b_k}$ in the reduced space of 3 PCs.

We set the number of endpoints using elbow criteria on the fraction of variance in the local cellular composition explained by a different number of endpoints. When varying the number of endpoints $p$, the number $d$ of PCs used for fitting the simplex was always $d = p - 1$ because $p - 1$ dimensions are generally needed to describe a simplex with $p$ endpoints.

## Assessing the robustness of niches to sampling intensity

In order to robustly identify niches while optimizing computation time, we performed an error analysis as a function of the sampling intensity—defined as the ratio of the total area of sites to the tissue area

—to test how deeply tissues need to be sampled so as to control for niche cellular composition error.

To minimize the sampling error, we first over-sampled the tissue by collecting a number of sites such that the total area covered 1000% of the tissue area. Over-sampling the tissue minimizes the sampling error because, even when sampling at 100% intensity, random positioning of sites may leave certain tissue areas uncovered by any site. Sites sampled at 1000% intensity were used to determine reference niches for the MIBI dataset of Keren et al.[21].

We then sampled sites such that the total area covered 300%, 100%, 30%, 10%, 3%, and 1% of the tissue area. At each sampling intensity, sites were sampled 100 times and niches were computed, producing 100 sets of four niches per sampling intensity. The niche estimation error was computed as the RMSE to the reference niches in terms of cellular composition. We plotted the root mean squared error averaged over the 100 repeats at each sampling intensity (Supplementary Fig. 5a).

A 30% sampling intensity—which we used in analyzing the MIBI data of Keren et al.[21]—identified niches with small enough an error to robustly characterize the biology of each niche (Supplementary Fig. 5b) while speeding up computations. If computation time is not an issue, we recommend sampling a number of sites equivalent to 100% of the tissue area or more, as the error is slightly smaller compared to sampling 30%.

## Classifying tumor samples into mixed vs compartmentalized using niche weights

To associate NIPMAP's niche segmentation with the previously proposed mixed vs compartmentalized classification of tumor architecture of Keren et al.[21], we sorted samples according to the contribution of tumor-immune interfaces relative to the total prevalence of immune niches, following the methodology described by Keren et al.[21]. More specifically, for each sample, the NIPMAP mixing score $m$ was computed as

$$m = \frac{\langle \alpha_3(\alpha_1 + \alpha_2) \rangle}{\langle \alpha_1 \rangle + \langle \alpha_2 \rangle}, \qquad (3)$$

where $\alpha_1$, $\alpha_2$, $\alpha_3$ represent the weight of the TLS, inflammatory, and cancer niches, respectively, at a given site, and averaging is performed over sites. The NIPMAP mixing score matched the mixed vs compartmentalized classification of Keren et al. for 37 out of the 40 samples (Supplementary Fig. 3a). We then tested whether the findings of Keren et al. on the association between mixed vs compartmentalized samples (reported in Fig. 5B, E, H of the original study) and the immuno-signaling environment could be reproduced using the NIPMAP mixing score. All three associations reported by Keren et al. could be reproduced using the NIPMAP mixing score (Supplementary Fig. 3b). Cold samples were excluded from the analysis, following the exclusion criteria of Keren et al.

## Comparing the spatial variation in cellular composition captured by different numbers of community ecology- and clustering-based niches

We compared how NIPMAP and clustering captured spatial variation in the composition of $m$ cell types across 4000 sites sampled from 40 triple-negative breast tumors analyzed by MIBI.

Iterating over the number of niches ($p = 2, \ldots, 17$ niches), NIPMAP was performed using $p - 1$ PCs $U \in \mathbb{R}^{m, p-1}$ to find $p$ niches $\boldsymbol{b_j} \in \mathbb{R}^{p-1} j = 1, \ldots, p$. We collect all niches $\boldsymbol{b_j}$ into a matrix $B \in \mathbb{R}^{p-1, p}$.

The percentage of variance in cellular composition explained by NIPMAP niches was computed as follows. For each site $k$, we compute the niche weights $\boldsymbol{\alpha_k} \in \mathbb{R}^p$ that best fit the site's cellular composition $\boldsymbol{c_k} - \boldsymbol{c_0} \simeq UB\boldsymbol{\alpha_k}$, where $\boldsymbol{c_0}$ is the average cellular composition of sites (we performed centered PCA). The difference between the site's best-fitted

cellular composition $UB\boldsymbol{\alpha}_k + \boldsymbol{c}_0$ and the observed site composition $\boldsymbol{c}_k$ is defined as the error $\boldsymbol{\epsilon}_k = \boldsymbol{c}_k - \boldsymbol{c}_0 - UB\boldsymbol{\alpha}_k$. 100% − the ratio of the squared error to the total sum of squares of site cellular is the fraction of explained variance by the niches, $100\% - \sum_{i,k}\epsilon_{ik}^2 / \sum_{i,k}(c_{ik} - c_{i0})^2$, where $i = 1, \ldots, m$ represents the cell type.

To determine how k-means clustering captures the spatial variation in cellular composition, we perform clustering to find $p$ niches $\boldsymbol{b}_j \in \mathbb{R}^m$ in the cellular abundance of our 4000 sites. The fraction of variance explained by $p$ niches—the clusters—was computed as $100\%$ − the ratio of the within-clusters sum of squares to the total sum of squares of the site composition data, $100\% - \sum_{i,k}(c_{ik} - b_{i\nu(k)})^2 / \sum_{i,k}(c_{ik} - c_{i0})^2$, where $v(k)$ is the cluster $j$ to which site $k$ belongs.

We note that NIPMAP's simplex model requires slightly more free parameters compared to k-means clustering, which helps increase the percentage of explained variance. We note that the goal of this analysis is not to find the best trade-off between the number of parameters and goodness of fit, but rather to identify a data structure that summarizes phenotypic spatial architecture in a concise fashion, using a small number of niches that fit well within the cognitive limitations of the humans.

## Power analysis of the probability to capture a rare niche

To study the power of NIPMAP to capture a rare niche, we simulated tissue data in which the prevalence of one niche varied while the prevalence of the remaining niches was set to be equal to each other. As a rare niche is expected to be more difficult to identify with little tissue data, we also varied the amount of tissue data in the simulation.

An existing approach to simulate tissue data[59] requires spatial co-occurrence statistics of cells of different types as an input. Tuning co-occurrence statistics so as to (a) specify the number and cellular composition of niches and (b) vary the abundance of a specific niche while keeping the other niches constant is not trivial. To address this, we designed a tissue simulation approach that can accommodate these two requirements, as follows.

We first simulated the spatial distribution of niche weights $\boldsymbol{\alpha}(x, y)$, where $\alpha_i$ is the weight of niche $i$ at position $(x, y)$ of the tissue, and $\sum_i \alpha_i = 1$. Simulated tissues should show continuous regions in which a given niche dominates, with smooth spatial transition into the contiguous niche. Drawing upon classical mathematical models of spatial patterns[60], we reasoned that a reaction-diffusion system in which niches compete locally with each other and diffusion enforces smoothness of niche weights in space could simulate realistic spatial distributions of niche weights.

Experimenting with different reaction-diffusion systems lead to the following equation:

$$\frac{d\alpha_i(x,y)}{dt} = \beta\alpha_i\left(\frac{\alpha_i^4}{\alpha_i^5 + K^5} - \frac{1}{2K}\right) + D\left(\frac{\partial^2\alpha_i}{\partial x^2} + \frac{\partial^2\alpha_i}{\partial y^2}\right) \qquad (4)$$

where we set $\beta = 1$/day.

The positive term in the first bracket represents cooperative logistic growth. When the weight $\alpha_i$ of niche $i$ is close to 0, there is near-zero niche growth. There is a step-like increase in growth as $\alpha_i$ approaches $K$, which rapidly saturates around 1, due to the high exponent of $\alpha_i$ (power of 5). By setting $K = 1/n$ where $n$ is the number of niches, we can thus establish growth dynamics in which only one niche wins at each location $(x, y)$.

The negative term in the first bracket of the equation prevents niche weights from growing to infinity, by ensuring that there are two stable fixed points in the absence of diffusion (that is when $D := 0$): $\alpha_i = 0$—niche $i$ is absent at $(x, y)$—and $\alpha_i \simeq 2K$—niche $i$ is present at $(x, y)$. The second bracket adds diffusion, to enforce smooth variation in niche weights with respect to space. We set the spatial domain in both $x$ and $y$ to $[0, L]$ with $L = 800\,\mu m$, the size of a MIBI image, using periodic boundary conditions.

We simulated this system numerically on a $50 \times 50$ lattice, that is $dx = dy = L/50 = 16\,\mu m$ until convergence using R's ode.2D solver from the deSolve library. At convergence, the $\alpha$s were normalized to sum up to 1 at each position $(x, y)$. We used $n = 4$ niches due to practical relevance to our re-analysis of the MIBI data of Keren et al.[21]. Setting $D = 40\,\mu m^2$/day produced niches whose spatial architecture resembles that of MIBI data (Supplementary Fig. 7a).

We define the prevalence of niche $i$ as $L^{-2}\int\alpha_i(x, y)dxdy$. To simulate tissues in which one niche is more rare—less prevalent—than the others, we altered the initial condition $\boldsymbol{\alpha}^0(x, y)$. We randomly initialized $\boldsymbol{\alpha}^0(x, y)$ so that, at each $(x, y)$, the weight of one niche was 1 and the weight of all other niches was 0. To simulate tissues in which niche 1 was less prevalent than other niches, we varied the probability $f$ that $\alpha_1(x, y)$ was 1. The probability that $\alpha_i(x, y) = 1$ for the other niches $i \neq 1$ was set to be equal. This resulted in initial conditions $\boldsymbol{\alpha}^0(x, y)$ in which the prevalence of niche 1 was $f$ and the prevalence of niche 2, 3 and 4 was $(1 - f)/(n - 1)$.

As expected, tissues represented by the initial condition $\boldsymbol{\alpha}^0(x, y)$ were unrealistic and unstructured: niches showed no spatial contiguity as niche weights varied abruptly from 0 to 1 from one location of the lattice to the next. Simulating the reaction-diffusion dynamics of $\boldsymbol{\alpha}(x, y)$ defined by Equation (4) to convergence caused niches to compete locally and laterally in space and thereby to establish contiguous areas in which a given niche dominated (Supplementary Fig. 7a).

Varying $f$ in the initial condition $\boldsymbol{\alpha}^0(x, y)$ from 0.04 to 0.25 in 9 logarithmic steps and simulating the system to convergence generated 9 tissues $\boldsymbol{\alpha}(x, y)$ in the form of matrices $50\times50\times n$ in which the prevalence of niche 1 ranged from 0.24% to 30.4% (Supplementary Fig. 7a).

From these nine simulated tissues $\boldsymbol{\alpha}(x, y)$, we simulated multiplex histology data. Multiplex histology data has the form of a table $X$ with $n_s$ rows representing sites and columns representing the local abundance of the different cell types.

To simulate this table $X$ for a given niche prevalence and a given number of MIBI images $n_i$ (amount of tissue data), we first computed the number of sites $n_s$ needed to cover the area of the $n_i$ images, $n_s = n_i L^2/(\pi r^2)$, where $r = 25\,\mu m$ is the radius of the site. Simulating sites from more tissue images is equivalent to collecting more sites from a given tissue image in the present scenario that different tissue images are made of the same niches.

We positioned $n_s$ sites in the tissue by sampling their position $(x, y)$ from a uniform distribution between 0 and $L$. For each site, we determined the local niche weights $\boldsymbol{\alpha}(x, y)$ by linear interpolation, to generate a matrix $A$ of sites (rows) × niches (columns). We simulated local cellular densities as $X = BA$, where $B$ is the matrix of niche cellular composition, whose rows represent cell types and columns represent niches. For realism, the cellular composition of the niches $B = (\boldsymbol{b_1}, \boldsymbol{b_2}, \boldsymbol{b_3}, \boldsymbol{b_4})$ was set to the four niches and 17 cell types of the MIBI data of Keren et al.[21].

We passed $X$ to NIPMAP to estimate 4 niches $\hat{\boldsymbol{b}}_i, i = 1, \ldots, 4$. We computed the RMSE $\epsilon$ on the estimated cellular composition of niche 1 from

$$\epsilon^2 = \frac{1}{17}\min_i ||\hat{\boldsymbol{b}}_i - \boldsymbol{b_1}||^2 \qquad (5)$$

where 17 is the number of cell types. The min operator ensures that the estimated niche closest to niche 1 is used to compute the error. This is necessary because niche indices are arbitrary in NIPMAP so that $\boldsymbol{b_1}$ doesn't necessarily match $\hat{\boldsymbol{b}}_1$.

We repeated this procedure 100 times for each niche prevalence and number of images $n_i$. Inspecting the estimated niche composition $\hat{B}$ as a function of the estimation error $\epsilon$ suggested that a threshold $\epsilon < 4.5 \times 10^{-4}$ distinguishes simulations in which niche 1 was accurately captured from simulations in which niche 1 failed to be captured. Thus, we estimated the probability to identify the rare niche 1 as the fraction of these 100 simulations for which $\epsilon < 4.5 \times 10^{-4}$ (Supplementary Fig. 7b).

## Identifying interface regions

Sites located at interface regions have high weights for more than one niche. Thus, to find sites at the interface of two niches, we compute the product of the niches' weight and look for sites where this product is high (Fig. 2f).

Interface regions can be defined in two ways. Under one definition, interfaces are found at the contact of two niches. To find these interfaces, we multiply the weights of pairs of niches. In this definition, interfaces are influenced by local cellular density. For example, a low concentration of immune cells next to cancer cells would not qualify as an immune-cancer interface because a low concentration denotes fibrotic regions (Supplementary Fig. 4f).

To identify interfaces between immune and cancer cells based only on cellular composition and independently of cellular density, we can exclude the contribution of the niche of low cellular density (here the fibrotic niche) by setting its weight to 0 at all sites and re-normalizing the weights of the other niches to sum up to 1. The benefit of this definition is that the interface regions identified through this process fit better with a visual impression, as the visual impression is guided more by cell types (colors) than cellular density (Fig. 2a, f).

## Processing and analysis of CyTOF data from Wagner et al.

We downloaded the summarized version of the CyTOF experiments of Wagner et al.[32]. The data table contains cellular proportions of cell types identified by 73 markers in 144 breast tumor samples. The cohort included 144 female patients, aged 29-93 (mean 62.8 years) and 1 male patient. The sample from the male patient was excluded from the analysis.

Cellular composition was organized in hierarchies, for example, the proportion of live cells among all cells, the proportion of cells of the M (myeloid) cluster among live cells, the proportion of M1 cells among the M cluster, and so on.

Wagner et al.[32] assigned cell types—tumor-associated macrophages, CD4+ T regulatory cells—to cell clusters (leaf nodes)—M01, T01, and so on. We took over these cellular assignments from Fig. 2D-L of Wagner et al.[32] in order to compute the relative composition of each tumor in terms of 12 cell types, chosen to be as similar as possible to the cell types profiled by Keren et al.[21]: cancer cells, fibroblasts, endothelial cells, CD4-T cells, CD8T cells, NK cells, dendritic cells, macrophages, B cells, plasma B cells, healthy tissue, other immune cells. One sample contained less than 50% live cells and was thus removed, keeping 143 samples for the analysis.

We performed PCA, centered and unscaled using the ade4 package of the data analysis software R[20]. Unscaled PCA was used because all features have the same units (fractional abundance is unit-less). We explored other transformations such as scaling by the standard deviation and log-transformation. Scaling by the standard deviation destroyed much of the covariance structure expected from breast tumor biology, presumably by amplifying sampling noise in low-abundance cell types. Log-transformation resulted in similar niches to the ones presented in the present article but produced curved simplexes which require developing new algorithms in order to fit the simplex to the data.

## Showing that inter-patient variation in macroscopic cellular architecture of tumors is constrained by a simplex whose endpoints are the microscopic niches

If inter-patient variation in the macroscopic cellular architecture of tumors is explained by patient-specific usage of universal niches, inter-patient variation in tumor cellular composition must be constrained by a simplex whose endpoints are the microscopic niches.

To see why, let $\alpha_j(x)$ be the local weight of niche $j$ at location $x$ of the tumor. All the weights can be collected into a vector $\alpha$, with $\sum_j \alpha_j = 1$ and $\alpha_j > 0$. We collect the cellular composition of each niche into a matrix B whose entries $b_{ij}$ indicate the density of cell type $i$ in niche $j$, in units of inverse volume ($1/\mu m^3$). With this notation, the local cellular composition $c(x)$ at location $x$ of the tumor is

$$c(x) = B\alpha(x) \qquad (6)$$

The macroscopic cellular composition of the tumor is then obtained by integrating the microscopic cellular composition $c(x)$ over the tumor volume $V$

$$C = \frac{1}{V}\int_V B\alpha(x)dx = B\frac{1}{V}\int_V \alpha(x)dx = B\theta \qquad (7)$$

Here, one can show that all $\theta_j$ are positive and sum up to one. First, since $\alpha_i \geq 0$,

$$\theta_i = \frac{1}{V}\int_V \alpha_i(x)dx \geq 0. \qquad (8)$$

Second,

$$\sum_i \theta_i = \sum_i \frac{1}{V}\int_V \alpha_i(x)dx = \frac{1}{V}\int_V \sum_i \alpha_i(x)dx = \frac{1}{V}\int_V 1dx = 1. \qquad (9)$$

Therefore, the macroscopic cellular composition of tumor $C$ is the weighted average of the niches B. Macroscopic tumor composition must be bounded by a simplex.

**In silico dissection of healthy tissue.** Direct comparison of microscopic niches and macroscopic cellular composition data is not possible because the tumor samples of Wagner et al.[32] partially include healthy tissue from the tumor margin whereas healthy tissue was not imaged in the samples of Keren et al.[21]. To enable a comparison of the two datasets, we mathematically dissect healthy tissue out of the tumor samples.

After projection onto $n-1$ PCs $v_i$, a CyTOF tumor sample $C$ (vector of proportions of 12 shared cell types) can be written as

$$C - \mu = \sum_{i=1}^{3} u_i v_i + \sum_{i=4}^{n-1} u_i v_i \qquad (10)$$

where $\mu$ is the vector of average proportions of each cell type and $u_i$ represents the contribution of PC $i$ to sample $C$. We then rewrite the first term as a weighted average of four endpoints (cancer, immunity, healthy, fibrotic) computed by archetype analysis in the space of the first 3 PCs to obtain

$$C - \mu = \sum_{j=1}^{4} \theta_j b_j + \sum_{i=4}^{n-1} u_i v_i \qquad (11)$$

where $b_j$ is the $j$th endpoint. To dissect the healthy endpoint (endpoint 3), we remove its contribution from the weighted averages:

$$\vec{\gamma} = \left(\frac{\theta_1}{\theta_1 + \theta_2 + \theta_4}; \frac{\theta_2}{\theta_1 + \theta_2 + \theta_4}; \frac{\theta_4}{\theta_1 + \theta_2 + \theta_4}\right) \qquad (12)$$

Finally, we compute cellular proportions $\vec{C}_d$ after dissecting the healthy endpoint as

$$C_d = \sum_j \gamma_j b_j + \sum_{i=4}^{n-1} u_i v_i + \mu \qquad (13)$$

For 15 out of the 143 CyTOF samples, the weight of the healthy endpoint was >50%, suggesting that healthy tissue dominated the cellular composition of these samples. These samples were discarded from

further analysis because the lower weight of the non-healthy endpoints risked increasing the dissection error.

**Mapping cell types across two datasets.** The cell types profiled in the CyTOF data of Wagner et al.[32] and MIBI data of Keren et al.[21] overlap only partially: while some cell types are common to both datasets—CD8T cells for example—other cell types were either only profiled in one dataset, or profiled at a different level of specificity—all B cells vs distinguishing B and plasma cells, all CD4 cells vs distinguishing CD4 cells and Tregs. This creates a challenge in comparing the two datasets. To address this, we mapped cell types across the two datasets to compute cellular composition based on cell types common to both datasets.

To do so, we created an incidence matrix $G$ of dimensions $m \times n$ with $m$ cell types from MIBI data as rows and $n$ cell types from CyTOF as columns. The entries of the $G$ matrix are set to 0 if the corresponding cell types from the two datasets are different and to 1 if they are identical. A column or a row can have more than one 1 if the MIBI and CyTOF datasets differ in their granularity for the corresponding cell type. This incidence matrix can be represented as a bipartite graph (Supplementary Fig. 7c).

From $G$, we then derive two matrices $G_k$, $G_w$ that allow projecting cell proportions from the initial MIBI $X_k$ and CyTOF data $X_k$ (respectively) onto the shared sets of cell types $Y_k = X_k G_k$ and $Y_w = X_w G_w$ by matrix multiplication.

To compute the projection matrix $G_k$ for the MIBI data, we initialize $G_k := G$. We then sum up all the rows of $G$. Rows where the sum is larger than 1 represent MIBI cell types that map to multiple, more granular CyTOF cell types. For these rows, we keep all columns with 0s and the first column with 1 in order to keep only the least granular cell type of the two datasets. The kept column is then named according to the row. To compute the projection matrix $G_w$, we perform the same procedure, reversing rows and columns.

Applying this procedure associates cell types of the MIBI and CyTOF datasets as indicated in Table 2.

## Hierarchical clustering of cell phenotypes and spatial specificity of phenotypes

To compare phenotypic clustering to the spatial phenotypes identified by NIPMAP, hierarchical clustering was performed on the intensity of 18 phenotypic markers previously classified as functional markers as opposed to lineage markers[61].

Marker intensities were Z-scored within each cell type to facilitate the visualization of phenotypic clusters and assess marker significance. Hierarchical clustering was performed on Z-scored intensities of all 18 phenotypic markers, in 3 cell types (dendritic cells, NK cells, and neutrophils) using euclidean distance and Ward linkage. To serve as a well-controlled comparison to the 10 niches and interfaces found by NIPMAP, 10 phenotypic clusters were determined for each cell type by cutting the hierarchical clustering dendrogram at the height needed to split the dendrogram into 10 groups using R's cutree function (dendextend package).

To quantify how phenotypic heterogeneity associates with space, we tested how accurately each phenotypic cluster predicted the niche of a given cell. We considered that a given cell was located in a given niche if the weight of that niche was at least 0.5. By tabulating how often cluster membership matched niche location, we computed the sensitivity and specificity of each cluster in predicting the different niches.

Phenotypic clusters are identified without regard to the niche location of cells. We thus asked whether a combined analysis of niche location and phenotypic markers could identify better predictors of the niche location of cells. To do so, we trained linear predictors of the niche weight of each cell based on the intensity of all 18 phenotypic markers. Changing the cut-off on the predicted niche weight beyond which a cell was considered to localize in that niche, we computed how

**Table 2 | Cell types from the MIBI data of Keren et al.[21] and CyTOF data of Wagner et al.[32] were aligned to compare the two datasets**

| shared cell type | cell type from CyTOF data | cell type from MIBI data |
|---|---|---|
| DC | DC | DC |
| Endothelial | Endothelial cells | Endothelial |
| Macrophages | Macrophages | Macrophages |
| Mesenchymal-like | Other | Mesenchymal-like |
|  | Fibroblasts | Unidentified |
| CD4-T | CD4+T | CD4+T |
|  |  | Tregs |
| Cancer | Epithelial cells | Tumor |
|  |  | keratin-positive tumor |
| NK | NK | NK |
| B | B cells | B |
|  | Plasma cells |  |
| Other immune | Other immune | Neutrophils |
|  |  | Mono/Neu |
|  |  | CD3+T/CD4-T |
|  |  | DC/Mono |
|  |  | Other immune |
| CD8T | CD8+T | CD8+T |

different niche weight cut-offs achieved different sensitivities and specificities (ROC curves).

## Quantifying the niche weights of individual cells

To associate cell phenotypes and niches (see next section), the niche weights of each cell need to be determined.

The niche and interface weights are computed for all cells of the dataset, by centering sites on each cell of the dataset. The cellular composition $c$ of each site is determined and the contributions of the different niches to each site $\alpha$ is computed as described above (Eqn. 6) by solving the matrix equation $c = B\alpha$, with $B$, a matrix of the cellular composition of the different niches.

We solve for $\alpha$, $\sum \alpha_i = 1$ by quadratic programming using the *qpsolvers* Python library with the default solver "quadprog". Cells labeled as *Unidentified* were discarded from the analysis.

## Identifying niche-phenotype associations

To identify associations between phenotypic markers and niches in a given cell type, we iterate over markers, niches, and cell types.

For each marker-niche-cell type triplet, we compute Spearman's rank correlation $\rho$ between marker intensity and niche weight in individual cells of the dataset. Here, a niche can be an individual niche $i$ (quantified as $\alpha_i$) or an interface region between two niches $i$ and $j$ (quantified as $\alpha_i \alpha_j$). We only consider combinations of cell types and niches for which there was at least one example of a cell of that type located mostly in this niche ($\alpha_i > 1/2$). For interfaces, the maximal weight of each niche is 1/2 (not 1 as in niches): thus we only consider combinations of cell types and interfaces $i, j$ for which there was at least one example of a cell of that type with $\alpha_i \alpha_j > 1/2(1/2)^2 = 1/8$. In the heatmap visualization of niche-phenotypes associations, the $\rho$ correlation of the marker-niche-cell type triplets that do not meet this criteria is set to 0.

Statistical significance is quantified as a p-value using a two-sided asymptotic $t$ test approximation. The calculation is repeated for all combinations of phenotypic markers of all cell types, and all niches/interfaces.

To correct for multiple testing and focus analysis on the clearest niche-phenotype associations, we compute the false discovery rate[33]

and keep only q-values smaller than 0.01. We also require the Spearman correlation to be at least 0.3, a threshold beyond which visual intuition confirms niche-phenotype associations.

We considered higher-order interfaces between niches (three niches and more) but chose not to report them here because niche-phenotype associations were weaker compared to niches and their pairwise interfaces.

In the analysis of the MIBI data of Keren et al.[21], to prevent false positives in niche-phenotype associations due to spatial signal bleed-over of cancer cell markers to neighboring cells, we filtered out the Keratin 6 and beta-catenin markers in niches dominated by cancer cells, that is the cancer niche and interfaces with the cancer niche. We also filtered out the p53 marker in Tregs of the cancer niche specifically because overlaying the cell segmentation mask and the p53 signal suggested spatial spill-over in Tregs of the cancer niche (Supplementary Fig. 10). We kept the p53 marker in the analysis so as to potentially capture the p53 phenotype in other cell types and other niches, as p53 localized away from cell membranes in most cells (Supplementary Fig. 10), consistent with its known pattern of nuclear accumulation upon DNA damage and other stresses[62]. We were left with 55 cell type and phenotype markers with significant spatial associations.

To summarize the heatmap of niche-phenotype associations into a tissue architecture table (Table 1, Fig. 5e), we iterate over all niches and interfaces. For each of these, we collect cells with phenotypic associations with $q < 1\%$ and $\rho > 0.3$.

To reduce redundancy in reporting phenotypic associations and clarify the niche vs interface specificity of phenotypic markers, we remove phenotypic markers from a niche if that niche has an interface with a larger $\rho$. Conversely, we remove phenotypic markers from an interface if that interface borders a niche with a larger $\rho$.

In each niche/interface, we also report cell types without specific niche-phenotype associations that are robustly present in the niche. To identify these cell types, we consider the 1% sites with the highest weight for that niche. For each cell type, we then compute the mean and standard deviation in the abundance of that cell type. Cell types whose mean abundance is at least one standard deviation away from 0 abundance are reported in that niche.

### Processing and analysis of in situ sequencing data from Sountoulidis et al.

The pcw13 embryonic human lung dataset from Sountoulidis et al.[44] was communicated to us by the authors in the form of a table where rows represent the cells of the dataset and columns represent the x- and y-position of the cells, the cells' type, and RNA quantification (molecular counts) of 89 phenotypic markers. The data can be downloaded from the dedicated github repository (see Data and Code Availability). The data can be visualized interactively on TissUUmap[63].

The authors clustered single-cell gene expression profiles to determine the type of individual cells in the dataset. The 73 initial cell types were simplified into 32 cell types to facilitate interpreting the spatial architecture of the lung tissue (Supplementary Data 1).

We performed niche identification on the 100,006 cells from the dataset, using the same approach used for the 40 tumor samples of Keren et al.[21] above. To accommodate the larger tissue size (6500 μm × 6500 μm) compared to the MIBI data (800 μm × 800 μm), 20,000 sites were generated, so that the total area of sites represented 100% of the tissue area. Computation time was not a limiting factor in this dataset because we performed fewer follow-up analyses compared to the MIBI data of Keren et al.[21]. Niche-phenotype mapping was then performed on an 800 μm × 800 μm region of the tissue illustrated in Fig. 5a.

### Visualizing projection of sites on the faces of high-dimensional simplexes

To visualize how a simplex fits the cellular composition of sites when more than 3 dimensions are needed to capture site cellular composition, we visualize the distribution of sites onto the faces of the simplex.

To do so, we iterate through all the faces of the simplex, defined by all combinations of three endpoints $i$, $j$, $k$.

We exclude sites located farthest from the face, as the projection of sites located far from a face provides little insight regarding how well the simplex's face fits these sites. To do so, we only keep sites for which the combined weight of the endpoints that define the face is at least 50%.

From the coordinates of these endpoints in PC-space, we determine an orthonormal basis for the three endpoints of the face and position the endpoints on that basis: $i$ at (0, 0), endpoint $j$ at $(x_j, y_j)$ and endpoint $k$ at $(x_k, y_k)$. Then, the position $p$ of sites on the face is determined from niche weights $\alpha$ as $p = \alpha_i(0, 0) + \alpha_j(x_j, y_j) + \alpha_k(x_k, y_k)$.

### Statistics and reproducibility

Statistics are reported with $p$ value, sample size, and bilaterality in the context of their use in the article. For reproducibility, full R/python code, including random seed, input data files, and documentation can be found on the companion github repository of this manuscript[64].

### Reporting summary

Further information on research design is available in the Nature Portfolio Reporting Summary linked to this article.

## Data availability

The data to reproduce the analyses can be downloaded at https://github.com/jhausserlab/NIPMAP[64]. Source data are provided with this paper.

## Code availability

The NIPMAP software package and the code to reproduce the analyses can be downloaded at https://github.com/jhausserlab/NIPMAP[64].

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

## Acknowledgements

The authors thank members of the Hausser lab for critical discussion, and specifically Petter Säterskog, Antony Cougnoux, Guilhem Panneau. We also thank Leeat Keren, Sergio Salas, and Benjamin Towbin for discussions and input on this project. The authors acknowledge the support from the Swedish Cancer Fund (21 1731 Pj), the Swedish Research Council (2018-02530), SciLifeLab, and Karolinska Institutet (all to J.H.).

## Author contributions

J.H. conceived and supervised the research. F.L. prototyped the NIPMAP approach on MIBI data with input from D.A. A.E.M. performed the research, developed the NIPMAP code, and performed all MIBI and CyTOF analyses of the article. Z.K. analyzed the ISS data. L.G. and A.E. performed complementary analyses. J.H. wrote the manuscript with input from A.E.M., Z.K., L.G., and A.E. J.H. acquired funding. All authors edited or commented on the manuscript.

## Funding

## Competing interests

The authors declare no competing interests.
