## [Peer Review File · Nature Communications]

REVIEWERS' COMMENTS

Note: please be aware that the numbering/order of reviewers seen below may no longer match the numbering on the reports that we have sent you via email as went along.

Reviewer #1 (Remarks to the Author)

Recent advances in multiplex histology allow the gain of large amounts of information on single cells and the ability to link this information to their localization in the tissue. Of course, these novel techniques require complex data processing to reconstitute and interpret the cellular architecture of a tissue.

In this study, the authors use niche phenotype mapping (NIPMAP) to interpret such large data sets.

With the use of existing MIBI (protein) and in situ sequencing (RNA) data sets, and using capture sites of 25 μm (2-4 cells) they convincingly illustrated that NIPMAP might be a valuable tool for tissue reconstitution.

In addition, they have analyzed putative correlations between cellular phenotypes and the localization in specific niches. They identified that MHCII+ B cells are mainly in the TLS, MHC I+ tumor cells are mainly in the interface between cancer cells and inflammation niches, keratin 6 in DCs of the inflammatory niche, IDO + and PDL1+ neutrophils at the cancer/inflammatory interface and CD45RO+ macrophages are mainly in the inflammatory niche. Those observations are novel and of high interest to the field.

But especially for their conclusions on CD45RO (CD45RO expression on macrophages is associated with the inflammatory niche), some important data on biological aspects are missing. Most important, it is not clear whether in the used MIBI data, CD45RO can be clearly distinguished from other CD45 isoforms, especially since all CD45RO exons are also present in CD45RA, RB and RABC. Basically, there are 2 possibilities: 1) if CD45RO can be distinguished from other isoforms, a clear enrichment of CD45RO+ macrophages in inflammatory niches (compared to other CD45 isotypes on macrophages in other niches) should be clearly stated. 2) if CD45RO can't be clearly distinguished from other isotypes, there probably is an overall enrichment of CD45, which also could mean that these cells are mainly from hematopoietic origin (and other niches might have mainly macrophages from another origin as e.g. the fetal liver). This issue should be cleared in the manuscript.

Reviewer #2 (Remarks to the Author)

Review Summary:

I like the idea of looking at the edges of the simplex for "archetypical" spatial patterning. The method seems practical and useful for spatial omics data. The method could be strengthened with demonstrations of statistical power and robustness checks for the parameter used.

Method Summary:

NIPMAP is a method to identify repeated patterns of cell type organization, "niches", from spatial data. These niches can be used to analyze the architecture of a collection of spatial samples.

The input to NIPMAP is a set of cell centroid coordinates and an associated cluster type for each cells.

NIPMAP randomly selects 100 sampling sites for each spatial sample.

For each site, it estimates the local cell type composition using gaussian kernel density estimation. The best radius (length-scale) was found to be 25 μm .

Given the samples of cell type composition, PCA is performed into p dimensions. The latent embeddings are used as an input to archetypal analysis to find $p+1$ edges of a simplex, that become the "niches."

Given these niches, the embeddings can be represented as a combination of niches that define

each spatial locations composition. This serves as the basis for downstream analysis.

Major Comments:

1. To estimate the initial niches, you use a 100 samples per image. Given a length scale of 25 μm , heuristically NIPMAP sees an area of $100 * \pi * (25 \mu\text{m})^2$ out of a total of $(800 \mu\text{m})^2$ total area, which is about 30%. This should be enough to accurately sample the "niches", but it would be helpful to see an empirical example that this is sufficient sampling a.k.a. how does this compare to uniformly sampling across the entire image?
2. For the analysis of in situ data from Sountoulidis et. al. (4.11), it is never explicitly stated how many samples per specimen were used for estimation. Is it the same 100 as the breast tumor analysis? Is that enough given that the images are larger (in μm^2)?
3. For the breast tumor dataset, the cell type is determined experimentally from a marker. In the single cell RNA-seq, data however, the cell type is inferred from the expression levels. Are the "phenotypes" then also RNA-seq expression? If so, would that confound your niche-phenotype correlation results?
4. The quality of NIPMAP's results would seem to depend highly on the quality of the clustering. Would it be possible to demonstrate with the RNA-seq data how changing the clustering changes (or confirms) the learned associations? e.g. using the original 73 cells types or another level of hierarchical grouping
5. What is the statistical power of this method? Can it capture a "rare" niche? Can you use either simulations (e.g. Baker 2023, In silico tissue generation and power analysis for spatial omics) or a theoretical null model to determine how many examples of a niche are needed to detect it?

Minor Comments:

1. I had never seen an apostrophe as the thousands place separator, and I think the comma (e.g. 1,000 instead of 1'000) is more common.
2. What does "with a risk to obscure tissue architecture" mean? (introduction, paragraph 14)

Reviewer #3 (Remarks to the Author)

This authors address questions of automated tissue structure identification which is a very interesting topic since multiplexed imaging, the basis for such endeavors, is increasingly applied in science. The authors highlight the problem of clustering based on cellular composition if tissues are largely made up of mixed niches which will not allow the identification of the actual niches. Such limitations can be overcome by using approaches from ecology which is presented here. Overall, the approach seems reasonable, is supported by the data and the paper is well written given the complexity. Some aspects still need clarification and explanation but overall I would recommend publication in Nature Communication following revision.

Major points to address:

- The approach the authors describe relies on predefined cell types and will only be able to uncover niches based on those cell types. Could the authors envision an approach that would work on the marker intensity of cells as opposed to cell types? This is potentially also a suggestion for future work or something to address in the discussion.
- It would be nice if the authors could show robustness of the approach. E.g. For the breast cancer samples, they chose 4 ecology based communities (3 PCs) which explain 82% of the variance in cellular composition. What if they choose 6 PCs at 25 μm radius? Would the findings from figure 2 differ?
- In the original paper from Keren et al a mixing score was defined. If I understand correctly, then, mixed samples from the original paper should correspond to samples dominated by interfaces in the niche approach. Could the authors reproduce some of the findings from Kerent al? E.g. mixed tumors had increased ratios of PD1+ CD8 T cells / PD1+ CD4 T or, any of the differences shown in the original paper in Fig. 5B, E, H?
- In figure 3A and table 1 there are some beautiful observations, e.g. that HLA-I is upregulated in tumor cells mostly at the interface to inflammation which makes sense since those cells would be

cells exposed to inflammation (e.g. IFN γ) which upregulated the expression of HLA-I. While some of the explanations make sense, the expression of CD138 is correlated positively with the fibrotic niche weight in NK cells? What does that mean? CD138 is a marker for plasma cells and I am not aware of the expression in NK cells. Is this spatial spill over? Similarly, there is a positive correlation between p53 in Tregs in the cancer niche. Which I am also not aware of the meaning. Potentially some of these associations are confounded by spatial spill over which is something the community is well aware of and the authors mention in the methods that they remove Keratin+ niches from the analysis. Generally, I do see from the methods and the discussion, that the authors are aware of the problem of spatial spill over. However, some of the biological interpretations should be double checked on images with segmentation masks. The authors should highlight some of the actual segmentation masks and the expression of markers such as CD138, HLA-I and CD45RO with NK cells and Macrophages/DCs, respectively, to increase confidence that these markers are actually expressed in the cells and not in neighboring cells of the niche.

- Similarly as above, In Fig. 4 the authors only show data for DCs. However, many more cell types and markers exist and typically due to their shape the segmentation of myeloid cells is mostly imperfect and suffers from neighboring signal. The authors should show another cell type, for example from the lymphoid lineage.
- The authors should also clarify the methods section on how they actually calculated sensitivities and specificities for prediction of niches and which cut offs were used to plot the different phenotypic clusters below the ROC curves since overall the sensitivity for most of the phenotypic clusters plotted to predict niches is rather low.

Minor:

- Results should be written in past tense.
- Page 7: The authors state that CD4 T cells in the inflammatory niche do not express FOXP3 which probably they cannot since all FOXP3+ CD4 T cells should have been assigned to Tregs beforehand. I recommend removing the sentence and potentially Figure S1D.
- In Fig 2J the authors display patients 35, 4, 28 and all others. In the text they say that this figure should display patients 5 and 17. Please resolve this discrepancy.
- Throughout the manuscript the authors should refer to the precise relevant methods sections.
- In the methods section 4.4 it is unclear where the term Bak comes from. Assuming it is the endpoint of the simplex but should be clarified.
- For completeness and understanding, the authors should add a heatmap of marker expression, either aggregated across cell types or for single cells (sub-sampled), for cell types to clarify the expression of markers in each cell type to the supplements.
- The first sentence of Results paragraph 2.4 is somewhat odd. Please rewrite.
- On page 14 bottom the authors state the epithelium surrounds alveolar space as expected. However, I struggle to see this from Fig. 5B. I assume that this is mostly based on the finding that cells are seen between those niches, correct? Please clarify the main text for these findings and the vascular/alveolar space. Additionally, I recommend adding a concluding sentence on the last paragraph to finish the section.

RESPONSE TO REVIEWERS COMMENTS

Reviewer #1 (Remarks to the Author)

Recent advances in multiplex histology allow the gain of large amounts of information on single cells and the ability to link this information to their localization in the tissue. Of course, these novel techniques require complex data processing to reconstitute and interpret the cellular architecture of a tissue.

In this study, the authors use niche phenotype mapping (NIPMAP) to interpret such large data sets.

With the use of existing MIBI (protein) and in situ sequencing (RNA) data sets, and using capture sites of 25 μm (2-4 cells) they convincingly illustrated that NIPMAP might be a valuable tool for tissue reconstitution.

In addition, they have analyzed putative correlations between cellular phenotypes and the localization in specific niches. They identified that MHCII+ B cells are mainly in the TLS, MHC I+ tumor cells are mainly in the interface between cancer cells and inflammation niches, keratin 6 in DCs of the inflammatory niche, IDO + and PDL1+ neutrophils at the cancer/inflammatory interface and CD45RO+ macrophages are mainly in the inflammatory niche. Those observations are novel and of high interest to the field.

We thank the reviewer for this endorsement.

But especially for their conclusions on CD45RO (CD45RO expression on macrophages is associated with the inflammatory niche), some important data on biological aspects are missing. Most important, it is not clear whether in the used MIBI data, CD45RO can be clearly distinguished from other CD45 isoforms, especially since all CD45RO exons are also present in CD45RA, RB and RABC. Basically, there are 2 possibilities: 1) if CD45RO can be distinguished from other isoforms, a clear enrichment of CD45RO+ macrophages in inflammatory niches (compared to other CD45 isotypes on macrophages in other niches) should be clearly stated. 2) if CD45RO can't be clearly distinguished from other isotypes, there probably is an overall enrichment of CD45, which also could mean that these cells are mainly from hematopoietic origin (and other niches might have mainly macrophages from another origin as e.g. the fetal liver). This issue should be cleared in the manuscript.

We thank the reviewer for bringing up this point to our attention, as it allows us to clarify the biology of the CD45RO+ phenotype of tumor macrophages.

In the discussion, page 17, we now write:

“NIPMAP identifies novel spatial phenotypes, such as an association between CD45RO expression and the localization of macrophages in the inflammatory niche of triple-negative breast tumors. CD45 is a hematopoietic marker which can be expressed as different isoforms CD45RA, CD45RB and CD45RC depending on alternative splicing of its three exons A, B, and C. The CD45RO isoform lacks the A, B, C exons. The different isoforms are associated with

specific phenotypes, at least in the context of lymphocytes cells [30, 31]. Thus, quantifying the CD45 isoforms specifically is a prerequisite to interpret the observation that CD45RO+ macrophages localize in the inflammatory niche of triple-negative tumors. The existing literature suggests that the CD45RO antibody (UCHL1 clone, Biolegend) used in the MIBI study we re-analyzed here [21] is specific to the CD45RO isoform of CD45. The UCHL1 clone has been used to specifically quantify CD45RO since the 1980ies [48, 49, 50, 51]. An early flow cytometry study in T cells found that the abundance of CD45RA and CD45RO abundance — as quantified by UCHL1 — correlate negatively, consistent with the specificity of UCHL1 to the CD45RO isoform [49]. Profiling CD45RO by means of the UCHL1 clone alongside CD45RA (by means of the 2H4, F8-11-13 or HI100 clones) is routinely used to discriminate between naive and memory T cells [49, 51]. In addition, the risk of non-specific quantification of CD45 isoforms in macrophages is mitigated by western blot, flow cytometry and full-length RNA-seq observations that bone marrow–derived macrophages from the spleen, white adipose tissue, liver and the peritoneal cavity lack the CD45RA, RB and RC isoforms and thus specifically express the CD45RO isoform [36]. These observations suggest that macrophages specifically upregulate the CD45RO isoform in the inflammatory niche of triple-negative breast tumors.”

In the results, page 12, we now write: “The literature suggests that the CD45RO signal we analyze here is specific to the CD45RO isoform (Discussion).”

Reviewer #2 (Remarks to the Author)

Review Summary:

I like the idea of looking at the edges of the simplex for “archetypical” spatial patterning. The method seems practical and useful for spatial omics data. The method could be strengthened with demonstrations of statistical power and robustness checks for the parameter used.

We thank the reviewer for this endorsement.

Method Summary:

NIPMAP is a method to identify repeated patterns of cell type organization, “niches”, from spatial data. These niches can be used to analyze the architecture of a collection of spatial samples.

The input to NIPMAP is a set of cell centroid coordinates and an associated cluster type for each cells.

NIPMAP randomly selects 100 sampling sites for each spatial sample.

For each site, it estimates the local cell type composition using gaussian kernel density estimation. The best radius (length-scale) was found to be 25 um.

Given the samples of cell type composition, PCA is performed into p dimensions. The latent embeddings are used as an input to archetypal analysis to find $p+1$ edges of a simplex, that become the “niches.”

Given these niches, the embeddings can be represented as a combination of niches that define each spatial locations composition. This serves as the basis for downstream analysis.

Major Comments:

1. To estimate the initial niches, you use a 100 samples per image. Given a length scale of 25 μm , heuristically NIPMAP sees an area of $100 * \pi * (25 \mu\text{m})^2$ out of a total of $(800 \mu\text{m})^2$ total area, which is about 30%. This should be enough to accurately sample the “niches”, but it would be helpful to see an empirical example that this is sufficient sampling a.k.a. how does this compare to uniformly sampling across the entire image?

We thank the reviewer for inviting us to confirm that sampling 30% of the imaging area is sufficiently representative of tissue histology.

In the manuscript, page 7, we now write:

“We note that niches were determined by collecting 100 sites per sample, so that the total area covered by sites represents 30% of the image area. Such a sampling intensity is sufficient to accurately identify niches while speeding up computations (Fig. S1K-L, Methods 4.4)”

In the methods, page 21, we now write:

“In order to robustly identify niches while optimizing computation time, we performed an error analysis as a function of the sampling intensity — defined as the ratio of the total area of sites to the tissue area — to test how deeply tissues need to be sampled so as to control for niche cellular composition error.

To minimize the sampling error, we first over-sampled the tissue by collecting a number of sites such that the total area covered 1000% of the tissue area. Over-sampling the tissue minimizes the sampling error because, even when sampling at 100% intensity, random positioning of sites may leave certain tissue areas uncovered by any site. Sites sampled at 1000% intensity were used to determine reference niches for the MIBI dataset of Keren et al. [21].

We then sampled sites such that the total area covered 300%, 100%, 30%, 10%, 3% and 1% of the tissue area. At each sampling intensity, sites were sampled 100 times and niches were computed, producing 100 sets of four niches per sampling intensity. The niche estimation error was computed as the root mean squared error (RMSE) to the reference niches in terms of cellular composition. We plotted the root mean squared error averaged over the 100 repeats at each sampling intensity (Fig. S1K).

A 30% sampling intensity - which we used in analyzing the MIBI data of Keren et al. - identified niches with small enough an error to robustly characterize the biology of each niche (Fig. S1L) while speeding up computations. If computation time is not a consideration, we recommend sampling a number of sites equivalent to 100% of the tissue area or more, as the error is slightly smaller compared to sampling 30%.”

2. For the analysis of in situ data from Sountoulidis et. al. (4.11), it is never explicitly stated how many samples per specimen were used for estimation. Is it the same 100 as the breast tumor analysis? Is that enough given that the images are larger (in μm^2)?

We thank the reviewer for this chance to clarify our methods. In the methods on page 30, we now write:

“To accommodate the larger tissue size (6500 μm x 6500 μm) compared to the MIBI data (800 μm x 800 μm), 20,000 sites were generated, so that the total area of sites represented 100% of the tissue area. Computation time was not a limiting factor in this dataset because we performed less follow-up analyses compared to the MIBI data of Keren et al.”

3. For the breast tumor dataset, the cell type is determined experimentally from a marker. In the single cell RNA-seq, data however, the cell type is inferred from the expression levels. Are the “phenotypes” then also RNA-seq expression? If so, would that confound your niche-phenotype correlation results?

We thank the reviewer for this opportunity to address potential confounding between the niche identification and niche-phenotype mapping stages of our analysis. In the discussion, page 18, we now write:

“To interpret spatial phenotypes, markers need to be separated into (1) markers of cell type and (2) markers of phenotypes, for exclusive use during niche identification and niche-phenotype mapping respectively. This is because using a given marker both for cell typing and phenotyping would necessarily identify the phenotype in the niches where the cell type is present, thus biasing niche-phenotype mapping.

Both datasets used here comply with this separation. The MIBI data from Keren et al. used 17 lineage markers to define cell types and another 18 functional markers to identify phenotypes [21]. The ISS data of Sountoulidis et al. [43] used 72 lineage markers to identify cell types and profiled a distinct set of 75 genes from the WNT, SHH, NOTCH and RTK pathways [43] which we used here in niche-phenotyping mapping.”

4. The quality of NIPMAP’s results would seem to depend highly on the quality of the clustering. Would it be possible to demonstrate with the RNA-seq data how changing the clustering changes (or confirms) the learned associations? e.g. using the original 73 cell types or another level of hierarchical grouping

We now explore how changing the cell type clustering affects the learned associations.

In the results page 16, we now write:

“To test the robustness of the identified niches with respect to cell type granularity, we repeated niche identification using the 73 cell types proposed by Sountoulidis et al. instead of the 32 coarser-grained cell types of Fig. 5. Increasing the number of cell types, we find five niches with cellular composition and spatial distribution similar to those found with 32 cell types (Fig. S4F-G). This suggests that niches show a degree of robustness to cell type granularity.”

The discussion page 17 now reads:

“When performing single-cell analyses, a decision needs to be made regarding the granularity of cell types. For example, T cells could be lumped together with other immune cells or be assigned a more granular type such as Th17 CD4+ T cell. Alternatively, cell typing could have intermediate granularity - such as CD4 T cells - and complemented by phenotypes - such as Th17. This raises the question of how cell type granularity impacts niche identification. In general, one expects optimal niche identification when analyzing cell types at a range of intermediate granularities and underperformance when the cell types are insufficiently granular or too granular. This is because insufficient granularity can prevent observing the cell types that characterize a given niche: for example, a vascular niche characterized by pericytes and endothelial cells cannot be identified if cells are coarsely grouped into epithelial vs non-epithelial. Conversely, too granular cell types - for example if each cell has its own type - prevent identifying recurrent patterns in the local cell type composition of tissues to reveal its niches. In the context of the ISS data of Sountolidis et al., our findings suggest that the niches identified are robust to the exact number of cell types used.”

5. What is the statistical power of this method? Can it capture a “rare” niche? Can you use either simulations (e.g. Baker 2023, In silico tissue generation and power analysis for spatial omics) or a theoretical null model to determine how many examples of a niche are needed to detect it?

We have now explored the statistical power of NIPMAP to detect a rare niche by means of simulation.

The results page 8 now read:

“The number of identifiable niches depends on their prevalence and on the amount of available tissue data. A power analysis based on tissue simulations suggests that a single MIBI image is sufficient to capture a niche whose area represents 3% of the tissue area or more (Methods 4.7, Fig. S1O-P). The probability to capture a rare niche scales as the product of niche prevalence times data size, so that increasing the amount of data allows identifying rarer niches. For example, the tissue area of 6 MIBI images allows identification of niches that represent a fraction of percent of the tissue area (Fig. S1P).”

The new method section 4.7 pages 22-24 now reads:

“To study the power of NIPMAP to capture a rare niche, we simulated tissue data in which the prevalence of one niche varied while the prevalence of the remaining niches was set to be equal to each other. As a rare niche is expected to be more difficult to identify with little tissue data, we also varied the amount of tissue data in the simulation.

An existing approach to simulate tissue data [59] requires spatial co-occurrence statistics of cells of different types as an input. Tuning co-occurrence statistics so as to (a) specify the number and cellular composition of niches and (b) vary the abundance of a specific niche while keeping the other niches constant is not trivial. To address this, we designed a tissue simulation approach that can accommodate these two requirements, as follows.

We first simulated the spatial distribution of niche weights [...]"

We add two new supplementary figures S1O and S1P whose legends read:

“O. The spatial distribution of four niches (colors) was simulated using a 4-species reaction-diffusion partial differential equations model. The initial condition was varied so as to simulate tissues in which one niche was more rare than the three other niches.

P. The probability to capture a rare niche increases with niche prevalence (columns) and with the total tissue area available for niche-phenotype mapping (simulated MIBI images as rows).”

Minor Comments:

1. I had never seen an apostrophe as the thousands place separator, and I think the comma (e.g. 1,000 instead of 1'000) is more common.

We have now replaced apostrophes with commas as place separators in all the numbers of the manuscript (e.g. 1,000 instead of 1'000).

2. What does “with a risk to obscure tissue architecture” mean? (introduction, paragraph 14)

On page 4, we have now re-written this paragraph for clarity:

“The notion that the local cellular composition of tissues does not necessarily form clusters of cellular composition can help interpret multiplex histology data in two ways. First, when sites do not form clusters, many clusters can be needed to describe tissue architecture, potentially leading to an inflation of clusters of unclear histological significance that over-complicate our view of tissue architecture with little scientific benefit. Interpreting local cellular composition as a continuum defined by a parsimonious number of niches can help address this.”

Reviewer #3 (Remarks to the Author)

This authors address questions of automated tissue structure identification which is a very interesting topic since multiplexed imaging, the basis for such endeavors, is increasingly applied in science. The authors highlight the problem of clustering based on cellular composition if tissues are largely made up of mixed niches which will not allow the identification of the actual niches. Such limitations can be overcome by using approaches from ecology which is presented here. Overall, the approach seems reasonable, is supported by the data and the paper is well written given the complexity. Some aspects still need clarification and explanation but overall I would recommend publication in Nature Communication following revision.

We thank the reviewer for this endorsement.

Major points to address:

1. The approach the authors describe relies on predefined cell types and will only be able to uncover niches based on those cell types. Could the authors envision an approach that would

work on the marker intensity of cells as opposed to cell types? This is potentially also a suggestion for future work or something to address in the discussion.

We thank the reviewer for challenging us to envision an approach to niche identification that does not rely on predefined cell types. A preliminary theoretical and data-driven exploration analysis suggests that it is indeed possible to identify histological niches based on marker intensities without knowing (a) the number of cell types, (b) the type of each cell, (c) the markers expressed by each cell type, and potentially (d) without segmenting cells.

In the discussion, page 18, we now write:

“Similar to how community ecology defines ecological niches based on local species covariance, local covariance between cell types is exploited by NIPMAP to identify histological niches. This approach requires assigning a type to each cell based on marker intensities and prior knowledge, with two potential downsides. First, niches identifiable in this way could be limited by previously known cell types. Second, assigning types to individual cells from marker intensities is time-consuming and not guaranteed to be error-free due to segmentation errors, signal mis-attribution, non-specific antibody binding, auto-fluorescence, molecular exchanges between cells and more. To address this, it would be desirable for niche identification to be based not on the types of cell but instead on marker intensities of local tissue regions prior to segmentation into cells. Preliminary exploration of this question in the context of the MIBI data of Keren et al. suggests that niches can potentially be identified without assigning predefined types to cells, resulting in similar niches as cell type-based niche identification (Supplementary Note 4). Future research can further develop this methodology.”

We added two new supplementary figures (S6A-B) and a new supplementary note 4 which reads:

“NIPMAP requires assigning a type to each cell based on its marker intensity profiles and prior knowledge, with potential downsides in terms of biasing niches according to prior knowledge of cell types as well as time and efforts in segmenting individual cells and assigning types to them. To address this, one can ground niche identification in marker intensities of local tissue regions prior to segmenting and assigning types to individual cells. Doing so is expected to be feasible if each (potentially unknown) cell type expresses a specific set of markers in a given (potentially unknown) niche.

To see why, we note that cell-type-based niche identification rests on the hypothesis that spatial heterogeneity in a tissue arises because the local weight of niches α varies in space, so that cellular composition x at a given site can be written as a weighted average of the cellular composition of the different niches B ,

$$x = B\alpha,$$

with α_i the local weight of niche i and $\sum_i \alpha_i = 1$. Here B is a matrix of dimensions cell types \times niches with the cellular composition of each niche.

Collecting the intensity of markers expressed by cells of a given type into a matrix M of dimensions markers \times cell types, we can write the intensities y of the different markers integrated over all cells at a given site as

$$y = M B \alpha = Q \alpha,$$

where $Q := MB$ represents the integrated marker intensities of each niche. Thus, integrated marker intensities at multiple sites are expected to fall on a low-dimensional simplex in the space of marker intensities, the marker equivalent of the cellular composition simplex of Fig. 2. Multiplying the cellular composition simplex by the matrix M of cell-type-specific marker intensities transforms it into the marker intensity simplex. The endpoints of the marker intensity simplex represent the tissue niches. The implication is that performing archetype analysis on the integrated marker intensities at many sites should uncover tissue niches. Note that we do *not* need to know the number of cell types, the type of a given cell, nor the markers expressed by each cell type to estimate the marker composition of the different niches Q by archetype analysis. Since the analysis is done on marker intensities integrated over the cells present at the site, segmenting the tissue into cells is potentially optional. We tested this idea using the MIBI data of Keren et al. [1]. We used the same methodology as in the analyses of Fig. 2 except for one point: instead of counting the number of cells of each type weighted by a Gaussian kernel ($\sigma = 25\mu\text{m}$), we summed up the intensities of each of the 34 markers over the cells present at a given site, weighted by the Gaussian kernel. Projecting the sites on 3 PCs shows that sites arrange themselves as a simplex (Fig. S4A), consistent with expectations. Inferring the endpoints of the simplex by archetype analysis identifies four niches whose markers are reminiscent of the niches identified by analyzing cellular composition (Fig. S4B). One niche is characterized by high expression of CD20, a B cell marker, suggesting a TLS. The second niche features high expression of CD68, a macrophage marker, and of CD3, CD8 and CD4, suggesting an inflammatory niche. The third niche expresses different Keratin markers, suggesting a cancer niche. The fourth niche shows low expression of all markers and could thus represent the fibrotic/necrotic niche. These observations suggest that niche identification can potentially be performed without assigning predetermined types to cells and in unsegmented tissue samples.”

2. It would be nice if the authors could show robustness of the approach. E.g. For the breast cancer samples, they chose 4 ecology based communities (3 PCs) which explain 82% of the variance in cellular composition. What if they choose 6 PCs at $25\mu\text{m}$ radius? Would the findings from figure 2 differ?

The revised manuscript now explores how the character of individual niches changes as one varies the number of ecology-based niches. The results, page 8 now read:

“Decreasing or increasing the number of niches from 4 niches down to 2 niches or up to 7 niches causes niches to merge into more coarse-grained niches or split into increasingly fine-grained sub-niches (Fig. S1M-N). While we used four niches here to balance accuracy and conciseness, this balance can be tuned by adjusting the number of the niches up or down to zoom in or out on the complexity of tissue architecture.”

The supplementary figure S1 now features 6 new panels whose legends read: “M. Increasing the number of niches produces more fine-grained niches. Tissue sections show the niche segmentation of Sample 17 using 2-7 niches. For 4 niches, see Fig. S1B and Fig. 2G. A two-niche segmentation of the tissue finds a tumor and a stromal niche. Adding a niche stratifies the stromal niche into a B / T CD4 rich region (pink) and a stromal niche with lower cellular density (red). Adding a fourth niche splits the red niche into an inflammatory (red) and fibrotic/necrotic niche (black). Adding a fifth niche identifies a macrophage sub-niche (orange) within the

fibrotic/necrotic (black) niche. A sixth niche splits the inflammatory niche (red) into a T CD4-rich (red) and a T CD8-rich niche (brown). Adding a seventh niche highlights a sub-niche rich in other immune cells (green) in the fibrotic/necrotic niche (black). N. A tree represents the pattern of successive niche splits as the number of niches grows.“

3. In the original paper from Keren et al a mixing score was defined. If I understand correctly, then, mixed samples from the original paper should correspond to samples dominated by interfaces in the niche approach. Could the authors reproduce some of the findings from Keren et al? E.g. mixed tumors had increased ratios of PD1+ CD8 T cells / PD1+ CD4 T or, any of the differences shown in the original paper in Fig. 5B, E, H?

We thank the reviewer for this invitation to connect our approach to the methods and findings of Keren et al.. We have now confirmed the reviewer’s intuition that mixed samples from Keren et al. correspond to samples dominated by interfaces in the NIPMAP approach. In the results, page 7, we now write:

“Sorting samples by increasing prevalence of tumor-immune interfaces (Methods 4.5) recovered the mixed vs compartmentalized sample classification proposed by Keren et al. (Fig. S1C) as well as previously reported differences in the immuno-signaling environment of mixed vs compartmentalized samples (Fig. S1D).”

The methods page 21 now read:

“To associate NIPMAP’s niche segmentation with the previously proposed mixed vs compartmentalized classification of tumor architecture of Keren et al. [21], we sorted samples according to the contribution of tumor-immune interfaces relative to the total prevalence of immune niches, following the methodology described by Keren et al. [21]. More specifically, for each sample, the NIPMAP mixing score m was computed as

$$m = \frac{\langle \alpha_3 (\alpha_1 + \alpha_2) \rangle}{(\langle \alpha_1 \rangle + \langle \alpha_2 \rangle)}$$

where α_1 , α_2 , α_3 represent the weight of the TLS, inflammatory and cancer niches at a given site, and averaging is performed over sites. The NIPMAP mixing score matched the mixed vs compartmentalized classification of Keren et al. for 37 out of the 40 samples (Fig. S1C). We then tested whether the findings of Keren et al. on the association between mixed vs compartmentalized samples (reported in Fig. 5B, 5E and 5H of the original study) and the immuno-signaling environment could be reproduced using the NIPMAP mixing score. All three associations reported by Keren et al. could be reproduced using the NIPMAP mixing score (Fig. S1D). Cold samples were excluded from the analysis, following the exclusion criteria of Keren et al..”

4. In figure 3A and table 1 there are some beautiful observations, e.g. that HLA-I is upregulated in tumor cells mostly at the interface to inflammation which makes sense since those cells would be cells exposed to inflammation (e.g. IFN γ) which upregulated the expression of HLA-I. While some of the explanations make sense, the expression of CD138 is correlated positively with the fibrotic niche weight in NK cells? What does that mean? CD138 is a marker for plasma cells and I am not aware of the expression in NK cells. Is this spatial spill over? Similarly, there is a positive correlation between p53 in Tregs in the cancer niche. Which I am also not aware of the

meaning. Potentially some of these associations are confounded by spatial spill over which is something the community is well aware of and the authors mention in the methods that they remove Keratin+ niches from the analysis. Generally, I do see from the methods and the discussion, that the authors are aware of the problem of spatial spill over. However, some of the biological interpretations should be double checked on images with segmentation masks. The authors should highlight some of the actual segmentation masks and the expression of markers such as CD138, HLA-I and CD45RO with NK cells and Macrophages/DCs, respectively, to increase confidence that these markers are actually expressed in the cells and not in neighboring cells of the niche.

We thank the reviewer for inviting us to examine the spatial distribution of specific marker signal relative to segmented cells, in order to increase the confidence that markers associate with specific cell types. We have now added a new Figure S3 to illustrate marker signal and segmentation masks for the marker-cell-type-niche triplets suggested by the reviewer.

We find that CD45RO expression in macrophages is cell-specific.

So is CD45RO expression in DCs. In NK cells, we exclude spill-over expression of CD138 from neighboring cells, and suggest instead that CD138 is extracellular, in a soluble form, and that NK cells localize in CD138-rich areas of the fibrotic niche. This is consistent with previous reports of CD138 expression in breast and other other solid tumors, mainly by fibroblasts and tumors cells [Kind 2019, Palaiologou 2014], and that the extracellular domain of CD138 can be proteolytically cleaved and released into the extracellular compartment to regulate inflammation and fibrosis [Gopal 2020].

The p53 signal in Tregs of the cancer niche is likely due to spatial spill-over.

In cancer cells, the data rules out spill-over expression of HLA-I from neighboring cells and suggests instead association of HLA-I with cancer cells, either through endogenous expression by cancer cells or from extracellular contribution of HLA-I in soluble form.

On page 9, in the results section on niche-phenotype mapping, we now refer to the new Fig S3: “In keratin-positive tumor cells, the MHCI marker associated with the interface of cancer and inflammation niches (Fig. 3D, Fig. S3). This suggests that MHCI expression in tumor cells could determine the position of the cancer-inflammation interface. Alternatively, the proximity of the inflammatory niche could induce MHCI in neighboring cancer cells or secrete MHCI as a soluble form [36] (Fig. S3). Niche-phenotype mapping also highlighted unexpected spatial phenotypes. CD45RO+ macrophages and dendritic cells localized in the inflammatory niche (Fig. 3E, Fig. S3).”

On page 16, the discussion now reads:

“Errors in cellular segmentation and lateral signal spill-over can lead to mis-assigning cell types and phenotypes, potentially leading to false-positives or false-negatives during niche-phenotype mapping [46] (see the p53 marker in Fig. S3 for example). Even in perfectly segmented tissues, marker signal can be mis-attributed: a marker can associate with cells of a given type within a given niche not because cells of that type express the marker but instead because the marker is

systematically present in that niche, perhaps as a soluble form or as a constituent of the extracellular matrix (see the CD138 marker in Fig. S3 for example).

NIPMAP is designed as the final layer of the multiplex histology data processing stack. It does not attempt to correct cellular segmentation, cell type assignment or signal attribution errors: these issues need to be addressed in the corresponding layers.

These issues are recognized in the multiplex histology field and ongoing methodological research is seeking to address them [47, 48, 46, 49, 50]. The statistical methodology employed by NIPMAP provides a degree of robustness to segmentation, cell type and signal attribution errors because niche-phenotype associations are only captured if they occur systematically across cells. Despite that, and until a definitive methodological solution to cellular segmentation, cell type assignment or signal attribution errors is established, spatial phenotypes highlighted by NIPMAP need to be confirmed by overlaying cellular segmentation with the spatial signal distribution of the corresponding phenotypic marker (Fig. S3)."

Throughout the manuscript, we now refer to marker intensity instead of marker expression because the latter refers to endogenous expression of a given gene marker which is better assessed by RNA profiling whereas multiplex histology surveys the intensity of marker proteins.

The legend of new Fig. S3 reads:

"Overlaying marker signal and niche-specific cellular segmentation masks to validate spatial phenotypes suggested by niche-phenotype mapping.

Row 1: Visual inspection supports CD45RO expression by macrophages localized in the inflammatory niche (red). CD45RO signal localizes in membrane regions of macrophages, including in macrophages that contact neighboring cells with less or no CD45RO signal. The CD45RO signal in macrophages cannot be explained by spatial spill-over because marker signal is expected to spill from cells with higher marker abundance over to neighboring cells. That spatial signal spill-over is unlikely is confirmed by macrophages outside the inflammatory niche (blue): these macrophages have less CD45RO signal as expected, despite being in contact with a CD45RO+ cell.

Row 2: Illustration of CD45RO expression by DCs localized in the inflammatory niche. Shown is a dendritic cell (red) with more systematic CD45RO signal than neighboring cells. The CD45RO signal in that cell cannot be explained by spatial spill-over because marker signal is expected to spill from cells with higher marker abundance over to neighboring cells. That spatial signal spill-over is unlikely is confirmed by a DCs outside the inflammatory niche (blue): this cell has less CD45RO signal as expected, despite being in contact with a CD45RO+ cell.

Row 3: Visual inspection suggests that the CD138 signal in NK cells of the fibrotic niche is explained by a tendency of NK cells to localize in CD138-rich areas of the fibrotic niche. CD138, also known as Syndecan-1, is a membrane protein that binds growth factors, adhesion receptors, soluble small molecules, proteinases, and other ECM proteins [2]. The CD138 signal localizes in NK cells of the fibrotic niche with little contact with other cells, thus ruling out spatial spill-over. Most of CD138 signal appears to be extracellular. This observation is consistent with previous reports of CD138 expression in breast and other other solid tumors, mainly by fibroblasts and tumors cells [3, 4], and that the extracellular domain of CD138 can be

proteolytically cleaved and released into the extracellular compartment to regulate inflammation and fibrosis [2].”

Row 4: Visual inspection suggests that the p53 signal in Tregs of the cancer niche is likely due to spatial spill-over. Spatial spill-over is consistent with the localization of Tregs with partial p53 signal next to (cancer) cells with systemic p53 signal. A Treg is also observed in an area with apparent extra-cellular p53 signal.

Row 5: Visual inspection suggests two hypotheses for the HLA-I signal in cancer cells at the cancer-inflammatory interface. One hypothesis is that cancer cells at the cancer-inflammatory interface express HLA-I. An alternative hypothesis is that HLA-I signal in that region is of extra-cellular origin, imputable to the soluble form of HLA-I [5].”

On page 30, the methods now read:

“We also filtered out the p53 marker in Tregs of the cancer niche specifically because overlaying the cell segmentation mask and the p53 signal suggested spatial spill-over in Tregs of the cancer niche (Fig. S3). We kept the p53 marker in the analysis so as to potentially capture the p53 phenotype in other cell types and other niches, as p53 localized away from cell membranes in most cells (Fig. S3), consistent with its known pattern of nuclear accumulation upon DNA damage and other stresses [62].”

- Similarly as above, In Fig. 4 the authors only show data for DCs. However, many more cell types and markers exist and typically due to their shape the segmentation of myeloid cells is mostly imperfect and suffers from neighboring signal. The authors should show another cell type, for example from the lymphoid lineage.

We have now extended the dendritic cells analysis of Fig. 4 to a myeloid cell type (neutrophils) and a lymphoid cell type (NK cells). We also extended the analysis from the inflammatory niche to all four niches. The results — documented by new panels of Fig. S2 — confirm the findings made in dendritic cells: (a) markers highlighted by phenotypic clustering overlap but differ from markers identified by niche-phenotype mapping, (b) spatial markers found by niche-phenotype mapping are missed by phenotypic clustering, (c) in each niche, a phenotypic cluster predicts niche location as accurately as niche phenotype mapping, though with low sensitivity.

The results page 12 now read:

“We examined how tightly phenotypic clusters associate with spatial context. As an upper bound for how precisely the marker panel and marker quantification can position cells in space, we use a linear predictor of a cell’s niche from phenotypic marker intensities (area under the curve = 0.89 in predicting which DCs locate in the inflammatory niche, Fig. 4D).

In dendritic cells, we find that one phenotypic cluster predicts the inflammatory niche as accurately as the linear model (Fig. 4D). Other DC clusters predict the location of DCs in other niches (Fig. S2C). These observations generalize to other cell types (Fig. S2D-E).”

The legend of new supplementary Figures 2D-E reads:

“D-E. Observations in neutrophils and NK cells are consistent with the hypothesis that the spatial context of cells is a stronger determinant of phenotype than cell-autonomous effects: if

cell-autonomous factors dominated phenotypic heterogeneity, (spatially agnostic) phenotypic clusters would associate poorly with space. As with dendritic cells (see Fig. 4), (a) markers highlighted by phenotypic clustering overlap but differ from markers identified by niche-phenotype mapping, (b) some spatial markers found by niche-phenotype mapping are missed by phenotypic clustering, (c) in each niche, a phenotypic cluster predicts niche location as accurately as niche phenotype mapping, though with low sensitivity.

D. Top: in neutrophils, niche-phenotype mapping (left) and (spatially-agnostic) phenotypic clusters (right) identify common phenotypic markers — HLA-I, HLA-DR, phospho-S6. Some spatial markers are not found by phenotypic clustering (CD45RO). Some markers identified by phenotypic clustering don't associate with space (CD138). Below: predictions of niche from phenotypic markers based on a linear model (black curves) and phenotypic clustering (red dots).

E. Top: in NK cells, niche-phenotype mapping (left) and (spatially-agnostic) phenotypic clusters (right) identify common phenotypic markers — CD63, IDO, PD-L1. Some spatial markers are not found by phenotypic clustering (CD138). Below: predictions of niche from phenotypic markers based on a linear model (black curves) and phenotypic clustering (red dots)”

- The authors should also clarify the methods section on how they actually calculated sensitivities and specificities for prediction of niches and which cut offs were used to plot the different phenotypic clusters below the ROC curves since overall the sensitivity for most of the phenotypic clusters plotted to predict niches is rather low.

We thank the reviewer for the opportunity to clarify the methods of this part of the analysis.

On pages 27-28, the methods section now reads:

“Marker intensities were Z-scored within each cell type to facilitate the visualization of phenotypic clusters and assess marker significance. Hierarchical clustering was performed on Z-scored intensities of all 18 phenotypic markers, in 3 cell types (dendritic cells, NK cells, and neutrophils) using euclidean distance and Ward linkage. To serve as a well-controlled comparison to the 10 niches and interfaces found by NIPMAP, 10 phenotypic clusters were determined for each cell type by cutting the hierarchical clustering dendrogram at the height needed to split the dendrogram in 10 groups using R's cutree function (dendextend package). To quantify how phenotypic heterogeneity associates with space, we tested how accurately each phenotypic cluster predicted the niche of a given cell. We considered that a given cell was located in a given niche if the weight of that niche was at least 0.5. By tabulating how often cluster membership matched niche location, we computed the sensitivity and specificity of each cluster in predicting the different niches.

Phenotypic clusters are identified without regard to the niche location of cells. We thus asked whether a combined analysis of niche location and phenotypic markers could identify better predictors of the niche location of cells. To do so, we trained linear predictors of the niche weight of each cell based on the intensity of all 18 phenotypic markers. Changing the cut-off on the predicted niche weight beyond which a cell was considered to localize in that niche, we computed how different niche weight cut-offs achieved different sensitivities and specificities (ROC curves).”

Minor:

- Results should be written in past tense.

In all the sections of the main text, we now use the past tense whenever we describe analyses we performed. Reasoning and references to established facts stayed in the present, consistent with recommendations [English Communication for Scientists, <https://www.nature.com/scitable/ebooks/english-communication-for-scientists-14053993/126083980/>].

- Page 7: The authors state that CD4 T cells in the inflammatory niche do not express FOXP3 which probably they cannot since all FOXP3+ CD4 T cells should have been assigned to Tregs beforehand. I recommend removing the sentence and potentially Figure S1D.

We thank the reviewer for this opportunity to better represent the methods of Keren et al. The results section page 7 now reads:

“A regulatory T cell phenotype can be excluded for the CD4 T cells in this niche as T regulatory cells were assigned their own cell type based on co-expression of CD4 and FOXP3 [21] (Fig. S1G).”

- In Fig 2J the authors display patients 35, 4, 28 and all others. In the text they say that this figure should display patients 5 and 17. Please resolve this discrepancy.

We thank the reviewer for catching this inconsistency. The sentence on page 8 now reads: Certain tissue sections make use of all niches (for example patient 35 in Fig. 2J) while others use only a few niches (patient 4 for example, Fig. 2J).”

- Throughout the manuscript the authors should refer to the precise relevant methods sections.

We now refer to specific methods sections throughout the manuscript.

- In the methods section 4.4 it is unclear where the term Bak comes from. Assuming it is the endpoint of the simplex but should be clarified.

The corresponding methods section page 22 now defines B and α_k in two sentences that begin as:

“NIPMAP was performed using $p-1$ PCs U [...]”

and

“For each site k , we compute the niche weights [...]”

- For completeness and understanding, the authors should add a heatmap of marker expression, either aggregated across cell types or for single cells (sub-sampled), for cell types to clarify the expression of markers in each cell type to the supplements.

We now added a heatmap of marker expression aggregated across cell types as new Fig. S2A to which we refer on page 9 of the manuscript:

“To do so, we took advantage of single-cell, spatial measurements of 18 phenotypic markers also profiled by Keren et al. [21] alongside the 17 lineage markers used to determine cell types (Fig. S2A).”

- The first sentence of Results paragraph 2.4 is somewhat odd. Please rewrite.

We thank the reviewer for pointing out the typo. The sentence now reads: “NIPMAP can find use in exploring fundamental questions regarding the cellular and phenotypic architecture of tissues.”

- On page 14 bottom the authors state the epithelium surrounds alveolar space as expected. However, I struggle to see this from Fig. 5B. I assume that this is mostly based on the finding that cells are seen between those niches, correct? Please clarify the main text for these findings and the vascular/alveolar space. Additionally, I recommend adding a concluding sentence on the last paragraph to finish the section.

We have now revised the text and added a new Fig. 5D to address this point. In the results, page 16, we now write:

“In well-formed ducts, we observed that the epithelium separates the ductal space from the smooth muscle niche (Fig. 5B,D), as expected.”

The legend of the new Fig. 5D now reads:

“D. Niche-phenotype mapping recovers known spatial associations between niches, such as the epithelial niche separating the smooth muscle niche from ductal/alveolar space. Shown is a 2D projection of sites on the face of the 5-niche simplex defined by the ductal/alveolar, epithelial and smooth-muscle niches. The sequential organization of ductal, epithelial, and smooth muscle niches is reflected by a depletion of sites at the ductal-smooth muscle interface relative to the ductal-epithelial interface. Thus, from the ductal/alveolar space, we first encounter the epithelial niche. Beyond the epithelial niche, the smooth muscle niche increases in weight.”

On page 16, the last sentence of the section now reads: “Thus NIPMAP generalizes to spatial RNA profiling data and healthy tissues.”

REVIEWERS' COMMENTS

Reviewer #1 (Remarks to the Author):

The authors convincingly addressed my concerns. I support acceptance of the manuscript

Reviewer #2 (Remarks to the Author):

Summary:

NIPMAP is a method to identify repeated patterns of cells in spatial tissue measurement, based on concepts from ecology. In short:

1. sites are sampled from the tissue and the composition of (predetermined) cell types estimated with a gaussian kernel
 2. PCA is used to find the main axis of variance of cell type composition
 3. Archetype analysis is used to identify the edges of the pca simplex and identify niches
- Given these identified niches, the niches present in different parts of tissue can be characterized. Four main niches are found for a cancer dataset that can be used to further interrogate associations with other phenotypes.

Major Comments:

The authors did a very thorough job of answering previous suggestions regarding sampling the space and identifying rare niches. I'm glad to see that the method is able to identify rare niches well even in low data regimes.

One last suggestion is to include a section at the beginning of results that lays out the method's steps (independent of the data). As a methodological paper, I think methods-oriented readers would appreciate that writeup. (this could also free section 2.1 to focus on the biological results rather than a mix of methods explanation + calibration and results).

Outside of the writeup, it would be good to see a code snippet in the github README Quick start on the exact command to run a script on any dataset

Minor Comments:

Introduction 1:

"Up until the 1950ies" should be "Up until the 1950s"

Results 2.1:

Would "two assumptions" be more appropriate than "two notions"? (maybe two principles if you're convinced of their truth)

""We found that 8 and more" should be "We found that 8 or more"

Do you need to mention hyperspectral unmixing algorithms if they aren't being used in the paper?

"Examining the niches' cellular composition allows interpreting the biology of each niche" could be "allows us to interpret the biology of each niche"

"The probability to capture" should be "the probability of capturing"

Results 2.4

The caption for figure 4E is a bit unclear, are the first two lines from the data in 4A or examples of what those conditions would hypothetically look like?

Discussion 3:

"1980ies" -> "1980s"

"NIPMAP adds a principle to determine the number of and identity of ..." -> "NIPMAP determines the determine the number of and identity of ..."
That sentence structure could be reversed, with the "to cluster based methods ..." phrase coming after the "NIPMAP adds." for clarity.

"NIPMAP also adds a geometric principle..."  "NIPMAP also uses a geometric principle to ..."

Reviewer #3 (Remarks to the Author):

I thank the authors for their revision. My comments were addressed and I do not see any further need for revision.

RESPONSE TO REVIEWERS' COMMENTS

Reviewer #2 (Remarks to the Author)

Summary:

NIPMAP is a method to identify repeated patterns of cells in spatial tissue measurement, based on concepts from ecology. In short:

1. sites are sampled from the tissue and the composition of (predetermined) cell types estimated with a gaussian kernel
2. PCA is used to find the main axis of variance of cell type composition
3. Archetype analysis is used to identify the edges of the pca simplex and identify niches

Given these identified niches, the niches present in different parts of tissue can be characterized. Four main niches are found for a cancer dataset that can be used to further interrogate associations with other phenotypes.

Major Comments:

The authors did a very thorough job of answering previous suggestions regarding sampling the space and identifying rare niches. I'm glad to see that the method is able to identify rare niches well even in low data regimes.

We thank the reviewer for this endorsement.

One last suggestion is to include a section at the beginning of results that lays out the method's steps (independent of the data). As a methodological paper, I think methods-oriented readers would appreciate that writeup. (this could also free section 2.1 to focus on the biological results rather than a mix of methods explanation + calibration and results).

We thank the reviewer for this suggestion. Separating the writing into a data-independent methods vs application to data has merits benefits of clarity but at the cost of seeming arbitrary : it would prevent us from building the methodology constructively, layer after layer, thus decreasing the educational quality of the writing.

We thus prefer to conserve the present outline of the results section.

At the same time, we now outline the NIPMAP methodology in a new panel d of Fig. 1 whose legend reads :

“Niche-phenotype mapping uses ideas from community ecology to automatically segment tissues into niches and their interfaces. Based on this segmentation, the phenotypic architecture of the tissue is summarized and salient spatial phenotypes are identified.”

The methods section 4.1 now reads :

“NIPMAP methodology overview”

Sites are sampled from the tissue and their composition in terms of (predetermined) cell types estimated with a Gaussian kernel. The main co-variance axes of cellular composition are identified by principal components analysis. Archetype analysis is used to fit a simplex to site cellular composition and thereby identify histological niches. From these niches, the original tissue is spatially segmented into niches and interface regions. Niches and interface are associated to phenotypic markers by correlation analysis to (1) summarize tissue phenotypic architecture and (2) identify salient spatial phenotypes.”

Outside of the writeup, it would be good to see a code snippet in the github README Quick start on the exact command to run a script on any dataset

We have now included a code snippet in the github README and simplified the Quick Start section : <https://github.com/jhausserlab/NIPMAP/tree/main>

Minor Comments:

Introduction 1:

"Up until the 1950ies" should be "Up until the 1950s"

We have now corrected the text accordingly.

Results 2.1:

Would "two assumptions" be more appropriate than "two notions"? (maybe two principles if you're convinced of their truth)

The text now reads "two principles decrease the number of axes required to interpret tissue architecture."

"We found that 8 and more" should be "We found that 8 or more"

The text now reads: "We found that 8 or more PCs"

Do you need to mention hyperspectral unmixing algorithms if they aren't being used in the paper?

Removing references to hyperspectral unmixing algorithms can simplify the writing.

We choose to mention hyperspectral unmixing in order to connect the fields of multiplex histology and satellite imaging. This can help both fields because future advances in satellite imaging algorithms - processing more data, finding more niches from less data, tolerating outliers, etc.. - can benefit multiplex histology, and conversely. In addition, the inspiration for NIPMAP came from satellite imaging, and we deem it important to acknowledge this field's contribution here.

"Examining the niches' cellular composition allows interpreting the biology of each niche" could be "allows us to interpret the biology of each niche"

The text now reads "Examining the niches' cellular composition allows us to interpret the biology of each niche".

"The probability to capture" should be "the probability of capturing"

The text now reads "The probability of capturing".

Results 2.4

The caption for figure 4E is a bit unclear, are the first two lines from the data in 4A or examples of what those conditions would hypothetically look like?

We thank the reviewer for pointing out an improvement to the figure legend. The legend of panel 4e now reads:

"Squared correlations are expected to be higher for niches and lower for interfaces in a scenario where niches structure phenotypic architecture (left). Conversely, under the hypothesis that phenotypic architecture is structured by interfaces, squared correlations will be higher for interfaces and lower for niches (middle). The data shows that both niches and interfaces have high squared correlations and thus contribute to phenotypic architecture (right)."

Discussion 3:

"1980ies" -> "1980s"

We have now made this change.

"NIPMAP adds a principle to determine the number of and identity of ..." -> "NIPMAP determines the number of and identity of ..."

That sentence structure could be reversed, with the "to cluster based methods ..." phrase coming after the "NIPMAP adds." for clarity.

"NIPMAP also adds a geometric principle..."  "NIPMAP also uses a geometric principle to ..."

We thank the reviewer for this suggestion. We have now rephrased this paragraph to highlight NIPMAP's conceptual contributions, with the intention to promote their adoption in other approaches to multiplex histology:

"To cluster-based methods aimed at identifying discrete cellular structures in multiplex histology data such as community detection based on spatial cellular networks and cellular neighborhood analysis [Danenberg 2022, Schurch 2020], NIPMAP adds two principles. The first is that the number and identity of histological niches can be determined by exploiting the simplex geometry of local cellular composition. The second is that the simplex geometry provides a criterion to distinguish niches from interfaces and automatically identify interfaces without parameter tuning."

Reviewer #3 (Remarks to the Author):

I thank the authors for their revision. My comments were addressed and I do not see any further need for revision.

We thank the reviewer for this endorsement.